# Seeing the Arrow of Time in Large Multimodal Models

**Zihui Xue**     **Mi Luo**     **Kristen Grauman**
The University of Texas at Austin

## Abstract

The Arrow of Time (AoT)—time's irreversible flow shaping physical events—is fundamental to video comprehension, yet remains a significant challenge for modern large multimodal models (LMMs). Current LMMs struggle to perceive and utilize temporal directionality in video when responding to language queries, obstructing deeper temporal understanding. We tackle this deficiency by first providing a critical analysis of existing benchmarks and models. We then introduce ArrowRL, a reinforcement learning (RL)-based training strategy with an innovative reverse reward that instills AoT awareness by encouraging divergent video interpretations between forward and reversed visual frames. For rigorous evaluation, we additionally develop AoTBench, a new multi-faceted benchmark probing temporally challenging questions. Experiments show ArrowRL greatly advances temporal perception: it not only achieves substantial improvements on our challenging AoTBench but also boosts performance on standard video question answering (VQA) benchmarks (with peak accuracy gains reaching over 20% and 10% respectively). This validates ArrowRL's effectiveness and highlights the critical need for dedicated AoT understanding in LMMs.[1]

## 1 Introduction

Our world unfolds with a distinct rhythm, governed by time's relentless forward march. We watch cream swirl into coffee, smoke disperse, or a glass shatter—common events whose reversal would feel instantly unnatural, or physically impossible (Fig. 1 (a)). This Arrow of Time (AoT) concept is rooted in physical laws and shapes the causal structure of reality [31]. We humans navigate the world with an innate grasp of the AoT, effortlessly perceiving the flow of events and decoding temporal directionality for visual narratives. Crucially, this sensitivity goes beyond identifying physical irreversibility; it fundamentally involves understanding the sequential progression of events and interpreting the semantic meaning embedded within that progression. Equipping intelligent systems with this broader AoT sensitivity—capturing how events unfold over time—is essential for achieving genuine video understanding.

Early research on video representation learning has leveraged the inherent temporal dimension of video as a form of self-supervision, designing a variety of interesting pretext tasks such as forward/reverse sequence classification (an early form of AoT) [52, 66, 23], video order prediction [48, 17, 30, 32] and enforcing temporal alignment [19, 75]. Throughout these developments, temporal understanding is treated as an intrinsic, *vision-only* problem based on physical cues.

Today the landscape of video understanding is rapidly shifting with the rise of large multimodal models (LMMs) [39, 8], which link video perception with the generative and interactive capabilities of large language models (LLMs) via textual interfaces. This evolution motivates us to rethink AoT from a *language-aware* perspective. A video's semantic meaning is often intertwined with the AoT: consider Fig. 1 (b), where the forward and reversed videos yield opposite interpretations conveyed through language. While many efforts aim to improve the general temporal understanding of these

---

[1]Project webpage: https://vision.cs.utexas.edu/projects/SeeAoT.

39th Conference on Neural Information Processing Systems (NeurIPS 2025).

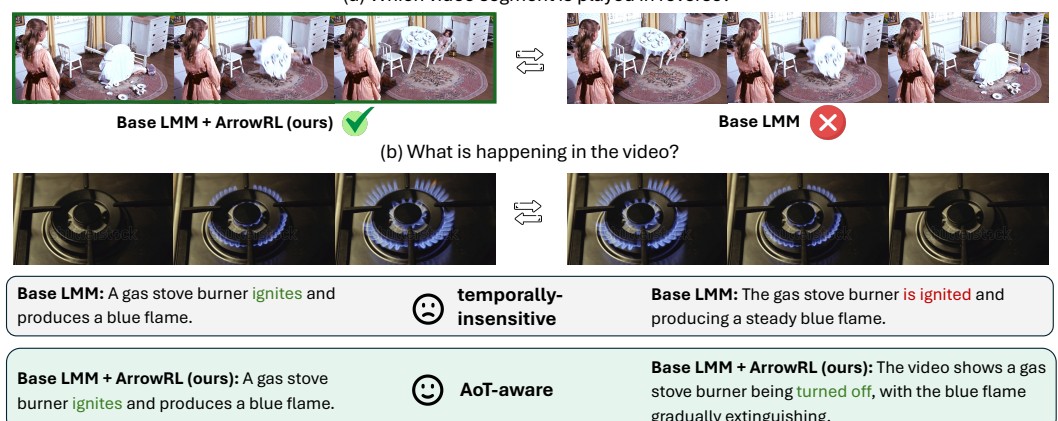

(a) Which video segment is played in reverse?

**Base LMM + ArrowRL (ours)** ✅          **Base LMM** ❌

(b) What is happening in the video?

**Base LMM:** A gas stove burner ignites and produces a blue flame.

☹️ **temporally-insensitive**

**Base LMM:** The gas stove burner is ignited and producing a steady blue flame.

**Base LMM + ArrowRL (ours):** A gas stove burner ignites and produces a blue flame.

🙂 **AoT-aware**

**Base LMM + ArrowRL (ours):** The video shows a gas stove burner being turned off, with the blue flame gradually extinguishing.

**Figure 1: Arrow of Time (AoT) perception challenges**, demonstrated by a strong representative base LMM (Qwen2.5-VL-7B [5]). (a) Basic visual directionality (forward vs. reverse), trivial for humans, often confounds these models. (b) Deeper temporal insensitivity is also observed, where LMMs often generate the same description (e.g., "ignite") for events with opposite semantics based on temporal direction. We propose ArrowRL to instill AoT awareness for LMM temporal perception.

powerful LMMs [85, 20, 77, 89, 38], we posit that AoT sensitivity is a fundamental yet overlooked component required for deeper understanding.

Consequently, we aim to instill this core temporal sensitivity, empowering LMMs to leverage directional information when generating language responses as appropriate. This capability is essential for advancing LMMs past superficial pattern recognition, as it would ground their internal world model [28, 27, 47] in the fundamental reality of temporal progression and causality, towards human-level reasoning.

Naturally, this leads us to ask: how capable are current LMMs in perceiving the AoT? Perhaps surprisingly, despite rapid advancements elsewhere, our findings indicate a critical gap. First, a profound temporal insensitivity is evident even in state-of-the-art LMMs. They fail on basic AoT-related tasks, such as distinguishing forward from backward video playback (Fig. 1 (a)), and ignore crucial semantic differences between forward and reverse sequences, erroneously providing identical descriptions (Fig. 1 (b)).[2] Second, broadening our investigation, we identify an alarming failure: on multiple standard video question-answering (VQA) benchmarks [70, 21, 51, 46, 9, 37, 36, 68], shuffling or reversing video frames often causes only slight or even no performance degradation for a leading representative LMM (see Fig. 2 and details in Supp. B.2).

The observed performance reflects a problematic interplay between intrinsic model capabilities and current benchmarks. First, on the modeling side, the observed failures clearly indicate a fundamental lack of temporal directionality understanding; this core perceptual capability for grasping causal-temporal dynamics remains underdeveloped in current LMMs. Second, on the benchmarking side, existing evaluations fail to adequately probe AoT sensitivity. While the single-frame bias issue [7, 33, 89, 15, 11, 69] (where one static frame suffices for answers) is widely studied, we identify a distinct critical limitation: many multi-frame questions still lack dependence on temporal order, allowing models to succeed even with shuffled or reversed video. We present a systematic study across eight existing VQA benchmarks to offer insights into this AoT insensitivity at both the benchmark and model levels, addressing a gap in prior evaluation studies.

Building on these insights, our work aims to empower LMMs to "see" the AoT in video. We first propose ArrowRL, a novel reinforcement learning (RL) algorithm based on Group Relative Policy Optimization (GRPO) [57]. The core idea is a unique reverse reward that promotes *divergence* between the model's forward and backward video interpretations, fostering AoT sensitivity for temporally demanding questions. In addition, to address benchmark inadequacies, as a secondary

---

[2]This is not a general failure to process reversed videos as out-of-distribution (OOD) inputs. See Supp. B.1 for a control experiment where LMMs excel on a temporally-insensitive task regardless of video direction.

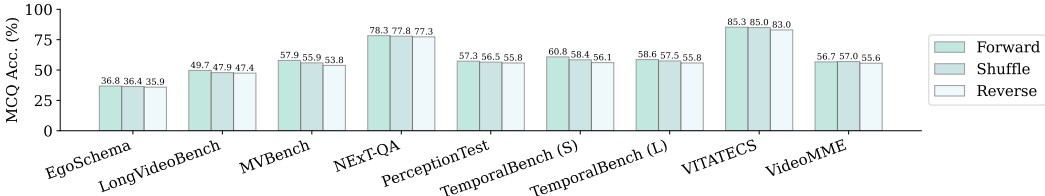

**Figure 2: Missing the AoT:** Multiple choice question (MCQ) accuracy of a strong representative LMM (LLaVA-OV-7B [34]) on standard VQA benchmarks [70, 21, 51, 46, 9, 37, 36, 68] when processing forward, shuffled, and reversed video sequences. S: short, L: long. The small or negligible performance drop across conditions highlights low temporal sensitivity, stemming from deficiencies in LMM capabilities and benchmark question design.

contribution, we develop AoTBench, a benchmark comprising three distinct tasks for rigorous evaluation of LMMs' AoT perception capabilities.

Our experiments demonstrate the efficacy of ArrowRL. Across three distinct base LMMs, it consistently boosts task performance on AoTBench, and, importantly, generalizes strongly to multiple standard VQA benchmarks [44, 80, 15] (e.g., +65.9% relative gain on [80]). These results underscore that improving temporal sensitivity in LMMs translates to broader performance gains, revealing the fundamental role of AoT awareness in achieving genuine temporal perception.

## 2 Related Work

**"Time" in Video**    Time has long been recognized as a valuable self-supervised signal in video [52, 66, 23, 48, 17, 32, 30, 19, 75]. Early works capitalize on this property within the visual domain, leveraging the arrow of time (AoT) by using forward/reverse sequence classification as a pretraining task, which benefits downstream action recognition [52, 66, 23]. Relatedly, random shuffling has been explored for action analysis [48, 53, 56]. Current multimodal vision foundation models [22], such as CLIP [54] and VideoCLIP [72], can benefit action understanding, driven by text-video contrastive alignment objectives. A few works [3, 65, 18] employ temporal reversal as a source of negative examples for aligning video and text encoders, yet the discussion remains specific to dual-encoder architectures and only addresses basic temporal concepts with before/after relations [3] or action antonyms [65, 18]. Critically, while the field has advanced to LMMs capable of free-form language-based interaction with video content [39, 8], we find that these models exhibit a surprising insensitivity to fundamental temporal directionality. This suggests their internal world models [28, 27, 47] may fail to incorporate the basics of real-world event progression and causality. While recent work investigates AoT in the context of text-only LLMs [50], and concurrent efforts explore it for physical reasoning [2], how to instill AoT awareness within LMMs remains a critical but largely unexplored problem.

**Large Multimodal Models (LMM)**    LLM capabilities have been extended to the visual domain through the development of LMMs [39, 8], which integrate a vision encoder with a powerful LLM, enabling them to perceive and reason about visual content. While early LMMs focus on images [42, 41, 87, 61], research on video-LMMs [12, 85, 89, 38, 34, 63, 5, 62, 43, 73, 14, 74] is rapidly progressing, including the development of high-quality video datasets [12, 85] and exploration of the architecture design space [89, 38, 43]. Our work is orthogonal to these data and architecture innovations. We investigate how a core understanding of the AoT can be instilled directly through training, offering a complementary path towards enhanced temporal perception in LMMs.

**Benchmarks with Temporal Sensitivity**    Evaluating the true temporal understanding of LMMs is challenging, even with many benchmarks available for video question answering [70, 21, 51, 46, 9, 44, 37, 36] and captioning [10, 62]. Recent studies [7, 33, 89, 11] reveal how questions on many popular video benchmarks can be answered correctly with minimal temporal context (single/no frame) due to static video content, non-temporal question design, or language queries that provide unintended hints. However, merely requiring multiple frames is insufficient for probing deep temporal understanding. Our work moves beyond the "single-frame bias" to address the overlooked issue that many multi-frame questions still lack sensitivity to temporal directionality. Inspired by preliminary experiments that explore frame shuffling on limited datasets [15, 69], our work provides, to our

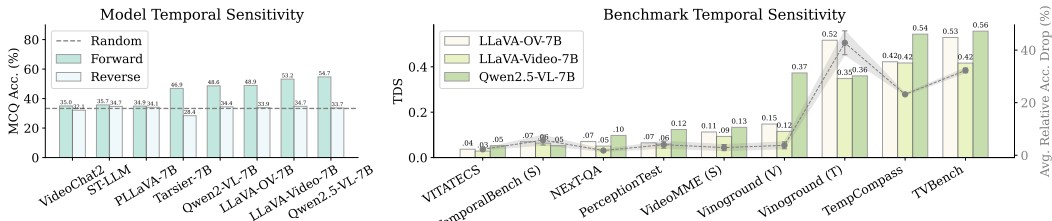

**Figure 3: Temporal Sensitivity Analysis**. (Left) **Model Sensitivity**: comparing MCQ accuracy on TVBench [15] for various LMMs [36, 43, 73, 62, 34, 85, 63, 5], on forward vs. reverse video sequences. LLaVA-OV-7B [34], LLaVA-Video-7B [85], and Qwen2.5-VL-7B [5] demonstrate highest accuracy and AoT sensitivity. (Right) **Benchmark Sensitivity**: comparing the proposed temporal divergence score (TDS) for various VQA benchmarks [37, 9, 70, 51, 21, 80, 44, 15], along with relative accuracy drop (mean ± std.) calculated using forward vs. reverse videos. Benchmarks with higher scores (Vinoground [80], TempCompass [44], TVBench [15]) are identified as temporally sensitive and better suited for evaluating temporal perception. S: Short, V: Video, T: Text.

knowledge, the first systematic and rigorous evaluation of AoT sensitivity across eight diverse VQA benchmarks. By comparing fine-grained metrics with forward, reversed, and shuffled video, we offer benchmark- and model-level insights currently lacking in the field.

**Reinforcement Learning to Enhance LMMs**   Post-training using reinforcement learning (RL) is increasingly employed to refine LLMs, aligning model responses with human preferences and enhancing complex reasoning abilities [49, 88, 55, 26, 57]. A family of successful algorithms, including PPO [88], DPO [55], online DPO [26], and GRPO [57], is now being actively extended to LMMs, although applications have mainly focused on image-LMMs [59, 78, 81, 84], with video comprehension being a newer frontier. While these efforts have concentrated on mitigating visual hallucination [83, 64, 79] or (concurrently) enabling structured reasoning [20, 13, 45] like Chain-of-Thought [67], our work investigates a distinct dimension: enhancing the model's fundamental temporal awareness, by leveraging AoT signals naturally embedded in videos.

## 3   Approach

Our preliminary experiments (Fig. 2) reveal a striking insensitivity to temporal directionality, suggesting deficiencies in both the benchmark design and/or LMMs' intrinsic ability to perceive the AoT. We systematically disentangle these intertwined factors, by first examining temporal sensitivity for existing benchmarks (Sec. 3.1), and then detailing our proposed approach to bestow AoT perception (Sec. 3.2) along with AoTBench, a new benchmark designed to evaluate this capability (Sec. 3.3).

### 3.1   Are Today's Benchmarks Time-Sensitive? Revisiting Current LMM Evaluation

To systematically study the inherent AoT sensitivity of existing benchmarks without costly human evaluation, we employ an automated strategy using LMMs to probe benchmark properties. To select suitable "evaluator" LMMs, we assess a pool of candidate models under 10B parameters (for computational feasibility)—VideoChat2 [36], ST-LLM [43], PLLaVA-7B [73], Tarsier-7B [62], Qwen2-VL-7B [63], LLaVA-OV-7B [34], LLaVA-Video-7B [85], Qwen2.5-VL-7B [5]—on TVBench [15]. TVBench is selected as its design explicitly requires a higher degree of temporal understanding [15]. Comparing accuracy for forward vs. reversed video on TVBench (Fig. 3 (left)) reveals leading models that exhibit both high overall accuracy and a significant difference (delta). Consequently, we select LLaVA-OV-7B [34], LLaVA-Video-7B [85], and Qwen2.5-VL-7B [5] for subsequent analysis.

We then evaluate the three selected LMMs across eight popular VQA benchmarks (VITATECS [37], TemporalBench [9], NExT-QA [70], PerceptionTest [51], VideoMME [21], Vinoground [80], TempCompass [44] and TVBench [15]).[3] To quantify benchmark and sample-level AoT sensitivity (facilitating later construction of AoTBench, c.f., Sec. 3.3), we propose the *temporal divergence score*

---

[3]Note that our analysis focuses on MCQ tasks for their popularity and clear evaluation structure, as opposed to open-ended QA or captioning tasks that require an additional LLM evaluator, introducing uncertainty.

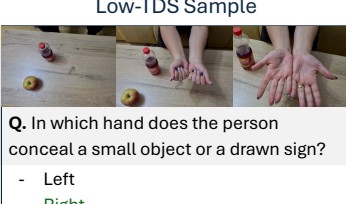

**Figure 4: Illustrative low vs. high TDS** VQA examples, sourced from PerceptionTest [51], NExT-QA [70] and TempCompass [44]. Samples with high TDS necessitate AoT reasoning, whereas the low-TDS sample can be solved without understanding video temporal progression.

(TDS). For an evaluator LMM $f$, consider an MCQ sample $i$, comprised of video $v_i$, language query $l_i$ and ground truth answer $y_i$ from $K_i$ options. Denote the reversed video as $\tilde{v}_i$. The MCQ setup instructs the model to respond only with the option's letter. The first-token probability distribution is therefore a direct measure of the model's confidence over the $K_i$ choices [6, 29, 40]. We compute the probability distribution for both forward and reversed video: $p_i = \text{FirstTokenProb}(f, v_i, l_i)$ and $\tilde{p}_i = \text{FirstTokenProb}(f, \tilde{v}_i, l_i)$, where $p_i, \tilde{p}_i \in \mathbb{R}^{K_i}$. To ensure our analysis focuses on instances genuinely requiring arrow of time (AoT) reasoning, we first filter out outlier samples where performance on the reversed video exceeds that of the forward one, as these cases typically indicate ambiguous or irrelevant temporal cues. The TDS for any remaining sample $i$ is defined using KL divergence: $\text{TDS}_i = \mathbb{D}_{\text{KL}}[p_i\|\tilde{p}_i]$. A high TDS signifies that the model's prediction is highly sensitive to the video's temporal direction—causing disagreement between forward and reverse responses—therefore suggesting the sample requires AoT understanding for a correct interpretation and answer.

Compared with prior efforts that examine accuracy delta (comparing full video vs. single-frame performance) [7, 33, 89, 15], which only reflects top-prediction flips, TDS is inherently more fine-grained by comparing the output probability distributions from *forward vs. reverse* video inputs via KL divergence—capturing shifts in confidence or the ranking of alternatives. This enables robust quantification of AoT sensitivity and allows for effective sorting of samples by AoT sensitivity.

Fig. 3 (right) presents the benchmark-level AoT sensitivity analysis, comparing TDS values. See Fig. 4 for low and high-TDS examples. We also report the coarser relative accuracy drop (averaged across evaluator LMMs) for reference. Our analysis reveals varying levels of AoT sensitivity across benchmarks: TVBench [15], Vinoground [80], and TempCompass [44] emerge as sensitive whereas benchmarks like VITATECS [37], TemporalBench [9], and NExT-QA [70] show lower sensitivity to video playback direction. These benchmark-level insights, derived from TDS, can aid future benchmark interpretation, and they directly influence our AoTBench design (Sec. 3.3).

### 3.2 ArrowRL: Learning the Arrow of Time

We now discuss our approach to equipping a given LMM $\pi_\theta$ with AoT understanding for enhanced temporal perception. Our core premise is that videos are inherently dynamic, and LMMs should capture this dynamic progression when responding. Specifically, given a question $q = (v, l)$ (composed of video $v$ and language query $l$) and a target response $o^\star$, drawn from the underlying data distribution $P(Q, O^\star)$ of a standard instruction tuning dataset, the goal is for $\pi_\theta$ to produce responses that align well with $o^\star$. To foster broad AoT sensitivity, our training utilizes a diverse suite of tasks—MCQ, open-ended QA, and video captioning (see Sec. 4 for the specific datasets used).

Given that modern LMMs are well-pretrained from extensive data, we follow advances in LLMs [49] and employ RL as a targeted post-training strategy. GRPO [57] is a leading RL algorithm that refines model outputs by comparing multiple responses generated for one prompt, adjusting probabilities based on relative scores within that specific group. In this way it provides finer control compared to using absolute rewards or response pairs and is thus well-suited for our goal of teaching AoT sensitivity. We adapt GRPO with a novel reward structure focused on temporal directionality, and term our approach ArrowRL. Fig. 5 provides an overview of the ArrowRL training process.

Given question $q$, let $\{o_i\}_i^G$ denote the set of $G$ candidate responses generated by $\pi_\theta$. We design a reward signal to rank these responses within the GRPO framework, integrating two complementary objectives: target fidelity and reverse reward.

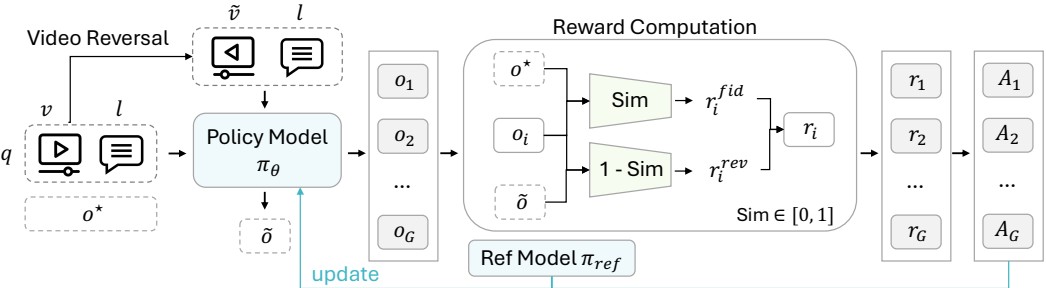

**Figure 5: Overview of our proposed ArrowRL framework**. Given input video $v$ and language prompt $l$, the policy LMM $\pi_\theta$ generates a group of candidate responses $\{o_i\}_{i=1}^G$. The reward calculation for response $o_i$ combines a fidelity reward ($r_i^{\text{fid}}$), encouraging similarity between $o_i$ and target $o^\star$, and a reverse reward ($r_i^{\text{rev}}$), enforcing temporal directional sensitivity between $o_i$ and a response $\tilde{o}$ generated from the reversed video $\tilde{v}$. These rewards $\{r_i\}_{i=1}^G$ are then normalized to obtain advantages $\{A_i\}_{i=1}^G$, which drive the GRPO optimization relative to the reference policy $\pi_{\text{ref}}$.

**Target Fidelity Reward** First, we define a fidelity reward encouraging alignment between each candidate response $o_i$ and the target response $o^\star$, $r_i^{\text{fid}} = \text{Similarity}(o_i, o^\star)$. The function $\text{Similarity}(\cdot, \cdot)$ returns a scalar between 0 and 1 and is calculated via accuracy matching (1.0 if $o_i = o^\star$, else 0.0) for MCQ tasks, or via an LLM-based similarity score for open-ended QA and captioning (see Supp. B.3 for the specific prompts and the LLM judge employed).

**Reverse Reward** Second, central to ArrowRL is the reverse reward. Rewarding only fidelity to the target response $o^\star$ (via $r_i^{\text{fid}}$) can be insufficient for capturing directional nuance—particularly if $o^\star$ lacks temporal progression details. Therefore, we utilize the reversed video (denoted by $\tilde{v}$) as a natural directional contrast signal, which often distinguishes itself from the forward version, to enforce AoT sensitivity. Intuitively, an AoT-sensitive model should yield different responses when observing the same video in different sequential order (though not without exceptions, which we handle below). Specifically, we maximize the dissimilarity between forward candidate responses $o_i$ and the reverse response $\tilde{o}$, i.e., the response produced by $\pi_\theta$ when given $\tilde{v}$. The reverse reward is thus defined as: $r_i^{\text{rev}} = 1 - \text{Similarity}(o_i, \tilde{o})$. This reward design discourages generic or static responses and compels attention to the dynamic, directional details conveyed by AoT in forward $v$. Note that in this process, the reversed video ($\tilde{v}$) is utilized solely as a signal for computing the reverse reward ($r_i^{\text{rev}}$); the training objective always pertains to generating correct and temporally-aware responses for the forward video ($v$), not to improving responses conditioned on the reversed video itself.

Fig. 6 illustrates this mechanism. A candidate response $o_i$ that incorrectly mirrors the reverse-conditioned response $\tilde{o}$ will receive a low $r_i^{\text{rev}}$. For instance (left example), both $o_2$ and $\tilde{o}$ describe "cooking beef" (where $\tilde{o}$'s description is contextually wrong for the forward video), thus $o_2$ is penalized via a lower reward. In the right example, given that the reversed video yields an $\tilde{o}$ of "picking up", candidate $o_2$ stating "placing...on" shows larger dissimilarity from $\tilde{o}$ than the more generic candidate $o_1$ stating "arranging". Thus, $o_2$ receives a higher reward as it better reflects the forward video's distinct temporal dynamics.

The two reward components work in tandem: the reverse reward compels the model to perceive AoT by enforcing divergence between forward and reverse responses, while the fidelity reward maintains relevance to the target response, preventing trivial outputs that merely differ from the reverse case. This complementary design encourages meaningful, temporally-aware responses. The final reward for ArrowRL on response $i$ is defined as: $r_i = r_i^{\text{fid}} + \alpha_i r_i^{\text{rev}}$, where $\alpha_i$ balances the two rewards.

A natural concern arises regarding the applicability of the reverse reward when its underlying premise—semantic distinctiveness between forward and reverse video—is not met (e.g. for static queries, non-dynamic or cyclic videos). We address this through training data selection (c.f., Sec. 4), adopting samples exhibiting high temporality, and also through a dynamic weighting scheme for the reverse reward's contribution $\alpha_i$. During training, we measure the similarity between the reverse response $\tilde{o}$ and the target one $o^\star$. If $\text{Similarity}(\tilde{o}, o^\star) > \gamma$ (where $\gamma$ is a threshold hyperparameter),

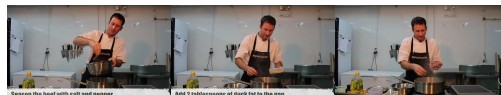
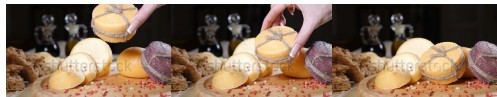

**Q.** After seasoning the beef, does the chef immediately start chopping vegetables?

**[Reverse response õ]** No, the chef does not immediately start chopping vegetables. Instead, he continues to cook the beef in the pan.

**[Candidate response $o_1$]** No, the chef doesn't immediately start chopping vegetables. After seasoning the beef, we see the chef preparing the pan with oil and duck fat. $r_1^{rev} = 0.45$

**[Candidate response $o_2$]** No, the chef starts cooking the beef in a pan instead of chopping vegetables. $r_2^{rev} = 0.15$

**Q.** Describe the video.

**[Reverse response õ]** The video shows a person picking up a piece of cheese from a wooden board.

**[Candidate response $o_1$]** The video displays a woman's hands arranging various types of cheese on a serving board. $r_1^{rev} = 0.6$

**[Candidate response $o_2$]** The video depicts a female hand reaching out and placing a piece of yellow cheese on a wooden table, which already has several other cheeses, as well as bread and berries. $r_2^{rev} = 0.8$

**Figure 6: Illustrating the reverse reward** ($r_i^{\text{rev}}$), designed to favor forward responses $o_i$ that are dissimilar to the response generated from the reversed video ($õ$), thus enforcing AoT sensitivity.

suggesting the sample is not highly AoT-sensitive, we disable the reverse reward for that sample by setting $\alpha_i = 0$. Otherwise, $\alpha_i$ takes a default hyperparameter value $\alpha$.

With this reward in hand, we proceed to the optimization step. For each question $q$ and the corresponding group of generated responses $\{o_i\}_i^G$, we compute the group reward $\mathbf{r} = \{r_i\}_i^G$, and then obtain advantage $\hat{A}_{i,t}$ as the normalized reward, i.e. $\hat{A}_{i,t} = \tilde{r}_i = \frac{r_i - \text{mean}(\mathbf{r})}{\text{std}(\mathbf{r})}$. The policy model is optimized by maximizing the following GRPO objective:

$$\mathcal{J}_{\text{GRPO}}(\theta) = \mathbb{E}_{(q,o^\star) \sim P(Q,O^\star),\{o_i\}_{i=1}^G \sim \pi_{\theta_{\text{old}}}(O|q)} \left[ \frac{1}{G} \sum_{i=1}^G \frac{1}{|o_i|} \sum_{t=1}^{|o_i|} \left\{ \min \left[ \frac{\pi_\theta(o_{i,t} \mid q, o_{i,<t})}{\pi_{\theta_{\text{old}}}(o_{i,t} \mid q, o_{i,<t})} \hat{A}_{i,t}, \right. \right. \right.$$

$$\left. \left. \left. \text{clip} \left( \frac{\pi_\theta(o_{i,t} \mid q, o_{i,<t})}{\pi_{\theta_{\text{old}}}(o_{i,t} \mid q, o_{i,<t})}, 1-\epsilon, 1+\epsilon \right) \hat{A}_{i,t} \right] - \beta \mathbb{D}_{\text{KL}} \left[ \pi_\theta \| \pi_{\text{ref}} \right] \right\} \right] \quad (1)$$

where $\epsilon$ and $\beta$ are hyperparameters controlling clipping and KL regularization respectively, $\pi_\theta$ is the current policy being optimized, $\pi_{\theta_{\text{old}}}$ is the policy from the previous iteration used for importance sampling and $\pi_{\text{ref}}$ is the reference model (set as the initial model checkpoint) used for regularization.

### 3.3 AoTBench: Addressing Temporal Evaluation Gaps in Video Understanding

Our preceding analysis (Sec. 3.1) and ArrowRL (Sec. 3.2) highlight the need for focused evaluation benchmarks. To this end, we propose AoTBench, the first dedicated benchmark to assess temporal direction sensitivity—a core component of robust video perception—through three distinct elements:

- **Sequence Direction Classification**. The task is to classify video playback as forward or reverse based on visual cues (Fig. 1 (a)), a capability with applications like video forensics, yet remains unexplored for LMMs. Following early vision work [66, 23], we adopt videos from Reverse-Film [66] and UCF101 [58], from which we manually select and verify a total of 613 videos demonstrating strong temporal properties suitable for this evaluation.

- **Directional Caption Matching**. This task probes the model's ability to connect video dynamics with corresponding textual descriptions. We target semantically directional video sequences where forward and reverse playback correspond to distinct textual descriptions (Fig. 1 (b)). Leveraging the RTime benchmark [18], which conveniently provides 2,000 such high-temporality videos alongside human captions, we formulate two complementary MCQ tasks: (1) video-to-text (V2T): choosing the correct caption (forward vs. reverse) for a given video, and (2) text-to-video (T2V): choosing the correct video sequence (forward vs. reverse) for a given caption.

- **AoT-sensitive VQA**. We curate temporally sensitive VQA samples from existing benchmarks. Using the per-sample TDS computed earlier, we select the top 200 high-TDS questions from each source benchmark, after removing easy samples unanimously answered correctly by all evaluator LMMs. This yields a subset of 1,800 VQA samples that necessitate robust AoT understanding and challenge current models' temporal perception capabilities. We illustrate the effectiveness of using TDS to isolate temporally challenging VQA samples in Fig. 4.

Table 1: **Benchmark results** comparing ArrowRL-enhanced LMMs with their base models, alongside leading open-source models of comparable scale and larger proprietary models (gray row). Cells marked with ⋆ indicate results reported in literature; others are reproduced using official code. AoTBench consists of three distinct tasks: (1) sequence direction classification (direc. cls.) on ReverseFilm (*RFilm*) and UCF101 (*UCF*), (2) directional caption matching (cap. match), and (3) AoT-sensitive VQA (AoT-VQA). ArrowRL consistently improves models' temporal perception ability across all three different base LMMs, leading to great performance gains on both AoTBench and existing video benchmarks.

| Model | AoTBench | | | | | Existing Temporal Benchmarks | | | | |
| | Direc. Cls. | | Cap. Match | | AoT- | Temp | TV | Vinoground | | |
| | *RFilm* | *UCF* | T2V | V2T | VQA | Comp. | Bench | Text | Video | Group |
|---|---|---|---|---|---|---|---|---|---|---|
| Random Chance | 50.0 | 50.0 | 50.0 | 50.0 | 39.3 | 44.3 | 33.3 | 25.0 | 25.0 | 16.7 |
| GPT-4o [1] | 52.8 | 54.0 | 56.5 | 69.5 | 67.8 | 74.8⋆ | 55.2⋆ | 54.0⋆ | 38.2⋆ | 24.6⋆ |
| Gemini-1.5-Pro [60] | 51.4 | 52.8 | 60.4 | 58.9 | 57.3 | - | 52.8⋆ | 35.8⋆ | 22.6⋆ | 10.2⋆ |
| Aria [35] | 50.0 | 51.6 | 56.4 | 57.5 | 55.9 | 73.6⋆ | 51.0⋆ | - | - | - |
| MiniCPM-V-2.6 [76] | 50.0 | 51.2 | 54.8 | 61.3 | 46.8 | 69.1⋆ | - | 32.6⋆ | 29.2⋆ | 11.2⋆ |
| InternLM-XC-2.5 [82] | 50.0 | 51.6 | 52.0 | 53.6 | 45.8 | 67.1⋆ | 51.6⋆ | 28.8⋆ | 27.8⋆ | 9.6⋆ |
| LLaVA-Video-7B [85] | 50.0 | 51.6 | 57.2 | 63.1 | 46.7 | 71.4 | 53.2 | 37.4 | 28.6 | 13.6 |
| Video-R1-7B [20] | 50.0 | 51.6 | 56.3 | 62.5 | 46.7 | 73.1 | 53.7 | 36.8 | 28.2 | 11.8 |
| LLaVA-OV-7B [34] | 50.0 | 51.6 | 56.3 | 62.4 | 46.2 | 69.6 | 48.9 | 42.2 | 29.0 | 14.8 |
| + ArrowRL | 54.2 | 57.4 | 57.9 | 66.1 | 53.2 | 70.7 | 49.9 | 43.4 | 31.2 | 17.6 |
| *(Gain)* | (+4.2) | (+5.8) | (+1.6) | (+3.7) | (+7.0) | (+1.1) | (+1.0) | (+1.2) | (+2.2) | (+2.8) |
| Qwen2-VL-7B [63] | 50.0 | 51.6 | 56.3 | 62.3 | 44.3 | 72.3 | 48.6 | 40.0 | 31.6 | 14.4 |
| + ArrowRL | 69.1 | 72.6 | 57.1 | 68.8 | 51.1 | 74.8 | 51.5 | 46.6 | 33.6 | 20.0 |
| *(Gain)* | (+19.1) | (+21.0) | (+0.8) | (+6.5) | (+6.8) | (+2.5) | (+2.9) | (+6.6) | (+2.0) | (+5.6) |
| Qwen2.5-VL-7B [5] | 50.0 | 51.6 | 53.4 | 66.6 | 49.6 | 73.8 | 54.7 | 46.2 | 31.4 | 16.4 |
| + ArrowRL | 51.4 | 54.8 | 55.6 | 69.6 | 58.8 | 75.5 | 56.2 | 48.8 | 42.4 | 27.2 |
| *(Gain)* | (+1.4) | (+3.2) | (+2.2) | (+3.0) | (+9.2) | (+1.7) | (+1.5) | (+2.6) | (+11.0) | (+10.8) |

## 4 Experiments

**Post-training Data**   Our ArrowRL training data comprises a comprehensive suite of tasks, with data selected or adapted for each to emphasize scenarios requiring AoT awareness: (1) MCQ tasks: we format sequence direction classification (Fig. 1 (a)) as MCQ, using 1.1K training videos selected from UCF101 [58] (following prior use of this dataset [23]); (2) Open-ended QA: we curate a high-temporality subset of LLaVA-Video-178K [85], filtering based on the perplexity difference between forward and reverse video to retain samples with great temporal sensitivity, totaling 11.8K samples; (3) Video Captioning: we employ the training set [18] of RTime, which provides high-temporality videos alongside distinct human captions for their forward and reverse versions, comprising 11.7K samples. See Supp. B.3 for data examples and prompts used.

**ArrowRL Implementation**   Our framework presents a straightforward and highly efficient approach by enabling direct application to pretrained LMMs without an intermediate supervised fine-tuning (SFT) step; we find ArrowRL alone is sufficient to instill AoT awareness. We demonstrate its broad applicability across **three leading base LMMs**: LLaVA-OV-7B [34], Qwen2-VL-7B [63], and Qwen2.5-VL-7B [5]. For efficiency and memory, we limit the maximum number of video frames to 16 during training. Hyperparameter $\alpha$ is set as 0.25 and $\gamma$ is set as 0.75. The response group size $G$ is set as 8. Training consists of 2000 RL steps on 6 NVIDIA GH200 GPUs.

**Evaluation Setup**   Our evaluation is designed to assess LMM temporal perception capabilities, using MCQ accuracy as the metric across multiple benchmarks. This includes our proposed AoTBench (Sec. 3.3), as well as three existing video benchmarks emphasizing temporal properties, as identified in Sec. 3.1: TempCompass [44], TVBench [15], and Vinoground [80]. We evaluate ArrowRL by comparing enhanced models to their respective base LMMs. Additionally, to situate these results within the broader landscape, we also report a diverse suite of leading LMMs as baselines. These include general-purpose models Aria [35] and MiniCPM-V-2.6 [76], video-focused models

**Table 2: Results on general video benchmarks** (base vs. Arrow-RL enhanced LLaVA-OV-7B). ArrowRL's specialization on temporal awareness does not degrade performance on these general video understanding tasks.

| Model | VideoMME (S) | NExT-QA | TemporalBench (S) | VITATECS | PerceptionTest |
|---|---|---|---|---|---|
| LLaVA-OV-7B [34] | 67.78 | 77.11 | 60.57 | 84.66 | 57.48 |
| + ArrowRL | 68.11 | 78.11 | 61.17 | 85.37 | 57.64 |

**Table 3: Ablation results** of ArrowRL. We report average performance across all columns of AoTBench tasks in Table 1. ArrowRL greatly outperforms the SFT baseline trained on the same data, demonstrating the effectiveness of our RL approach. In addition, using our curated high-temporality post-training data provides a performance gain, validating our data selection strategy.

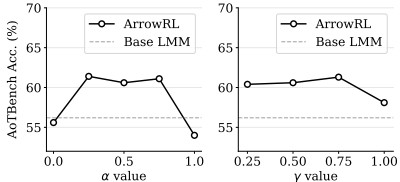

| Model | Training data | Acc. (%) |
|---|---|---|
| Qwen2.5-VL-7B [5] | - | 56.2 |
| + SFT | ArrowRL post-train set | 57.4 |
| + ArrowRL | LLaVA-Video-178K captions | 57.7 |
| + ArrowRL | RTime captions | 60.4 |
| + ArrowRL | ArrowRL post-train set | 61.4 |

**Figure 7: Hyperparameter analysis** of ArrowRL on AoTBench, using Qwen2.5-VL-7B as the base LMM. We identify an optimal setting at $\alpha = 0.25$ (reverse reward weight, left) and $\gamma = 0.75$ (dynamic weighting threshold, right), which effectively balances the fidelity and reverse reward.

LLaVA-Video [85], InternLM-XC-2.5 [82] (strong in fine-grained video understanding), and Video-R1 [20] (emphasizing video reasoning), along with two major proprietary (and much larger and more extensively trained) models GPT-4o [1] and Gemini-1.5-Pro [60].

**Main Results** As shown in Table 1, ArrowRL effectively enhances models' temporal perception capabilities across diverse settings. Results on AoTBench (left table) reveal critical deficiencies in current LMM AoT perception. Visually-focused tasks like sequence direction classification prove especially challenging: nearly all open-source models exhibit a universal failure mode: their responses are identical regardless of video input, resulting in consistent chance-level performance (50.0% on ReverseFilm and 51.6% on UCF101). Leading proprietary models (GPT-4o, Gemini-1.5-Pro), while slightly better, still perform far below satisfactory levels. This is in stark contrast to the proprietary models' absolute accuracy on other mainstream VQA tasks, where accuracy hovers around 70-85%. Our finding and the proposed AoTBench illuminate visual temporal bottlenecks. Importantly, this improved AoT awareness also translates to gains on existing, unfiltered temporal VQA benchmarks (right table), such as +10.8 group score (+65.9% relative gain) for Qwen2.5-VL-7B on Vinoground. These results underscore the promise of enhancing AoT perception for overall temporal perception. See Supp. C.1 for a follow-up discussion of the relative strengths of different base models and how they translate to ArrowRL's largest margins.

The most important outcome is ArrowRL's improvements over each base model, and, following that, its advantage over the open-source baselines—all of which are similar in scale at 10B parameters or less. The significantly larger proprietary models trained on much larger datasets (gray rows) are a useful high-level reference, but do not constitute a direct apples-to-apples comparison.

**Results on General Video Benchmarks** To ensure that specializing in AoT does not degrade general video understanding, we evaluate ArrowRL-enhanced LLaVA-OV-7B on five benchmarks previously identified as less temporally sensitive (see Fig. 3 right). As shown in Table 2, ArrowRL preserves or even slightly improves performance on these general benchmarks where AoT is not required, indicating it enhances temporal perception without sacrificing general video understanding.

**Ablations** Fig. 7 presents our hyperparameter analysis, confirming two key design choices. First, the reverse reward is essential: the left plot shows that removing it ($\alpha = 0$) degrades performance to below that of the base LMM. Second, the dynamic weighting mechanism is beneficial: the right plot shows that disabling it ($\gamma = 1$) yields less improvement than our dynamic approach, which can

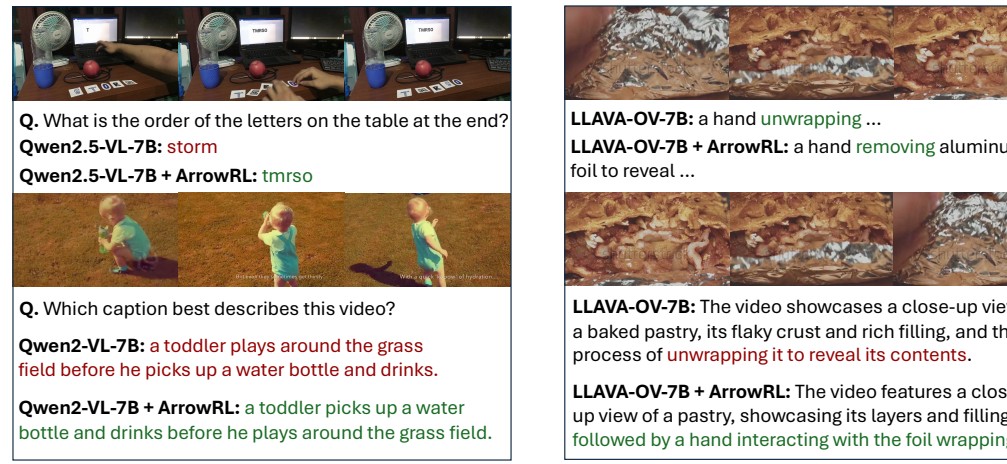

**Figure 8: Qualitative results** comparing base LMMs vs. Arrow-RL enhanced models. ArrowRL correctly answers AoT-sensitive VQA questions (left) and generates distinct, temporally accurate captions (right), while the base model fails to capture temporal progression in these examples. More examples and failure cases in Supp. C.2.

zero out the reward for non-AoT-sensitive samples. We further investigate different inference frame settings and find ArrowRL generalizes robustly despite its fixed-frame training. See Supp. C.1 for details.

**Qualitative Results**   Fig. 8 provides qualitative examples, comparing base LMMs against our ArrowRL enhancement. The VQA examples (left) demonstrate ArrowRL enabling base models to correctly answer temporally challenging questions. Similarly, when prompted to describe forward and reverse sequences (right), the base model provides the "unwrapping" description for both, failing to connect frames dynamically to understand the temporal progression and possibly influenced by language bias (favoring the more common "unwrapping"). Conversely, ArrowRL demonstrates clear temporal awareness by producing distinct and correct captions reflecting the temporal direction. See Supp. C.2 for more qualitatives, including failure cases.

**Limitations and Future Work**   While AoTBench grounds temporal directionality in language, we acknowledge that describing irreversible events remains inherently difficult in natural language. Future work could explore how learned representations encode time beyond language [4], and probe whether models develop specialized activations for time-reversed, physically implausible inputs [16]. For ArrowRL, important future directions include extending our methods to long videos (e.g., via keyframe selection) and exploring integration with temporal reasoning tasks that necessitate Chain-of-Thought reasoning. See Supp. D for a comprehensive discussion of limitations.

## 5   Conclusion

Our work underscores the crucial yet unexplored role of AoT understanding within LMMs. Towards this end, we introduce ArrowRL, an GRPO-based reinforcement learning algorithm with a novel reverse reward design to enhance temporal perception, and develop AoTBench, a comprehensive benchmark targeting temporal sensitivity. Great performance gains on both AoTBench and existing VQA benchmarks validate our approach and highlight the crucial role of AoT in advancing temporal understanding. We hope ArrowRL offers a promising path towards LMMs with true temporal understanding capabilities, and that AoTBench serves as a valuable tool for future research.

**Acknowledgements**   This research was supported in part by the UT Austin IFML NSF AI Institute, with compute on the Vista GPU Cluster through the Center for Generative AI (CGAI) and the Texas Advanced Computing Center (TACC). We would like to thank the anonymous reviewers for their thoughtful and constructive feedback, which provided valuable guidance in refining this work.

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

# Contents

## A  Supplementary Video

We invite readers to view the supplementary video available at `https://vision.cs.utexas.edu/projects/SeeAoT` for a visual demonstration of our work's overview and additional qualitative examples. Understanding how video content differs between forward and reverse playback—a key aspect of our study—is most effectively conveyed through video. Therefore, the supplementary video will offer readers a more intuitive grasp of how ArrowRL successfully enhances temporal sensitivity in LMM responses and how AoTBench effectively probes this crucial capability.

## B  Method Details

### B.1  Control Task to Disentangle OOD Effects

To verify that LMMs' poor performance on AoT tasks (observed in Fig. 1 of the main paper) is not simply due to reversed videos being treated as out-of-distribution (OOD) inputs, we design a control experiment inspired by [3]. Using videos and action labels from UCF101 [58], we formulate a binary-choice question with the prompt, "What action is being performed in this video?" The correct action is provided as one choice, and the incorrect option is randomly drawn from the set of all other action labels. We then evaluate several LMMs on both the original (forward) and time-reversed versions of these videos.

**Table 4:** We conduct binary action classification on UCF101 as a control task, to disentangle AoT insensitivity from out-of-distribution (OOD) effects. All LMMs achieve near-perfect accuracy regardless of video playback direction, suggesting that failures on AoT tasks are not due to a general inability to process reversed videos.

| Model | Binary Acc. (%) | |
| --- | --- | --- |
| | forward | reverse |
| GPT-4o [1] | 100.0 | 100.0 |
| Gemini-1.5-Pro [60] | 98.2 | 99.4 |
| LLaVA-OV-7B [34] | 99.2 | 99.4 |
| Qwen2.5-VL-7B [5] | 99.4 | 99.4 |

The results in Table 4 show that all models perform at near-perfect accuracy on this control task, irrespective of video direction. This contrasts sharply with their near-random performance on our AoT-sensitive sequence direction classification task (Table 1, UCF column). This analysis provides strong evidence that the failure of existing LMMs on AoT tasks is due to their inability to perceive temporal directionality, not a general failure to process reversed videos as OOD inputs.

## B.2 Temporal Sensitivity Analysis

The temporal sensitivity analysis, visualized in Fig. 2 of the main paper, is conducted using LLaVA-OV-7B [34] across several evaluation benchmarks: EgoSchema [46], LongVideoBench [68], MVBench [36], NExT-QA [70], PerceptionTest [14], TemporalBench [9], VITATECS [37], and VideoMME [21]. To ensure consistent and fair evaluation, we utilize *lmms-eval*[4] and a standardized setting of 16 input frames across three conditions: forward, reversed and shuffled video. We specifically select deterministic MCQ tasks to obviate the need for potentially unreliable third-party LLM evaluators and mitigate evaluation ambiguity.

Our subsequent analyses, including the development of AoTBench, focus on short videos. This approach allows for a targeted investigation of AoT awareness, separating it from complexities specific to long video processing, especially since current LMMs already struggle with shorter temporal sequences. Table 5 presents the forward, reversed, and shuffled frame performance (i.e., MCQ accuracy) for three selected LMMs (LLaVA-OV-7B [34], LLaVA-Video-7B [85], and Qwen2.5-VL-7B [5]) on several VQA benchmarks: VITATECS [37], TemporalBench [9], NExT-QA [70], PerceptionTest [51], VideoMME [21], Vinoground [80], TempCompass [44] and TVBench [15]. These results, which supplement the TDS-based benchmark sensitivity analysis in Fig. 3 of the main paper, further illustrates the great variance in temporal order sensitivity across benchmarks: some, such as Vinoground, TempCompass, and TVBench, demonstrate a stronger ability to probe this, whereas others, like VITATECS, show extreme insensitivity.

**Table 5:** Impact of video frame order manipulation (forward, reversed, shuffled) on MCQ Accuracy (%) for three LMMs across various VQA benchmarks. S: short, V: video, T: text.

| Benchmark | LLaVA-OV-7B | | | LLaVA-Video-7B | | | Qwen2.5-VL-7B | | |
|---|---|---|---|---|---|---|---|---|---|
| | forward | reverse | shuffled | forward | reverse | shuffled | forward | reverse | shuffled |
| VITATECS | 85.27 | 83.00 | 84.98 | 87.74 | 85.87 | 87.28 | 82.95 | 81.33 | 81.54 |
| TemporalBench (S) | 61.92 | 57.53 | 59.04 | 62.85 | 58.43 | 59.57 | 68.36 | 66.24 | 67.23 |
| NeXT-QA | 78.29 | 77.31 | 77.85 | 81.97 | 80.27 | 80.80 | 81.47 | 79.80 | 79.80 |
| PerceptionTest | 57.15 | 55.72 | 56.52 | 67.63 | 65.07 | 65.57 | 68.81 | 64.86 | 65.89 |
| VideoMME (S) | 70.89 | 69.22 | 68.89 | 76.33 | 72.89 | 73.67 | 72.33 | 71.00 | 69.56 |
| Vinoground (V) | 58.00 | 56.90 | 52.30 | 56.90 | 54.20 | 49.90 | 56.00 | 53.40 | 48.60 |
| Vinoground (T) | 68.20 | 35.80 | 53.80 | 65.90 | 36.80 | 51.80 | 61.10 | 38.70 | 49.50 |
| TempCompass | 69.78 | 53.34 | 62.16 | 72.09 | 55.29 | 63.24 | 73.88 | 57.15 | 65.21 |
| TVBench | 49.11 | 33.86 | 40.51 | 53.19 | 34.73 | 42.77 | 54.69 | 33.66 | 42.34 |

## B.3 ArrowRL

**Post-training Data** To enhance AoT understanding, our training data incorporates three core tasks. For MCQ-based sequence direction classification, we use selected videos from UCF101 [58]. For video captioning, we leverage the RTime dataset [18] with a varied set of 16 prompts. For open-ended QA, we employ original questions from LLaVA-NeXT-178K [85]. The prompts used for MCQ and captioning tasks are detailed below. For the MCQ-based sequence direction classification task, we follow [80] to concatenate the forward video and its reversed version—separated by a 2-second black frame—into a single video input, as LMMs typically process one video stream at a time.

---

[4] https://github.com/EvolvingLMMs-Lab/lmms-eval

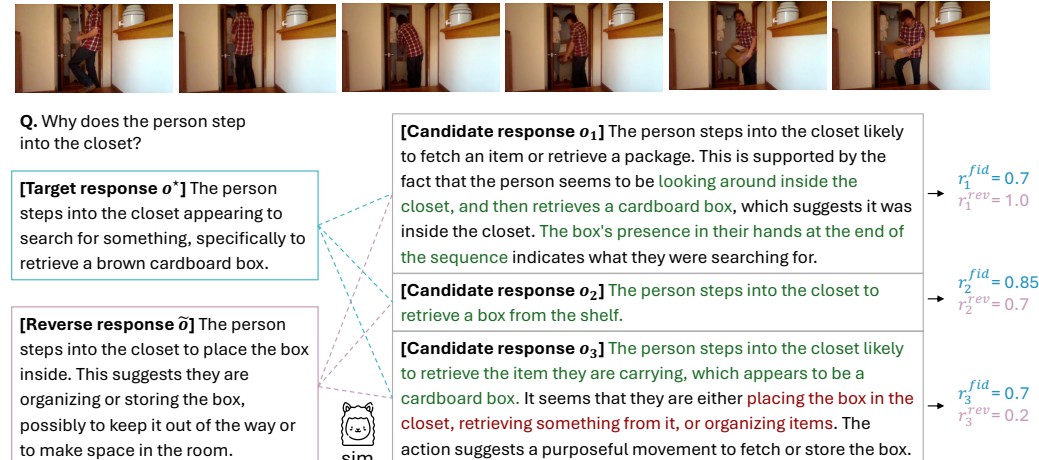

**Training Prompt I (MCQ-based Sequence Direction Classification)**

This video contains two segments showing the same action, separated by a 2-second black frame. One segment is played forwards, and the other is played in reverse. Which video segment is played in reverse?
A. The first segment (before the black frame)
B. The second segment (after the black frame)
Answer with the option's letter from the given choices directly.

**Training Prompt II (Video Captioning)**

Describe the following video in detail.
Provide a detailed description of the given video.
Give an elaborate explanation of the video you see.
Share a comprehensive rundown of the presented video.
Offer a thorough analysis of the video.
Explain the various aspects of the video before you.
Clarify the contents of the displayed video with great detail.
Characterize the video using a well-detailed description.
Break down the elements of the video in a detailed manner.
Walk through the important details of the video.
Portray the video with a rich, descriptive narrative.
Narrate the contents of the video with precision.
Analyze the video in a comprehensive and detailed manner.
Illustrate the video through a descriptive explanation.
Examine the video closely and share its details.
Write an exhaustive depiction of the given video.

**Figure 9:** An illustration of the reward calculation process for ArrowRL, using one VQA example from LLaVA-Video-178K [85]. An auxiliary LLM is employed to compute similarity scores between responses. While the fidelity reward $r_i^{\text{fid}}$ ensures candidate responses $o_i$ align with the target $o^\star$, the reverse reward $r_i^{\text{rev}}$ cultivates AoT sensitivity by using dissimilarity from the reverse response $\tilde{o}$ as its signal. Consequently, temporally correct and sensitive responses that diverge from $\tilde{o}$ (like $o_1$) are favored, while those that misunderstand AoT (like $o_3$, highlighted red) are penalized via a lower reward.

**Reward Calculation** Fig. 9 demonstrates ArrowRL's reward calculation using a VQA example that necessitates causal-temporal reasoning. Here, the fidelity reward ($r_i^{\text{fid}}$) alone might be insufficient;

for instance, candidates $o_1$ and $o_3$ both exhibit high similarity to the target $o^\star$. The reverse reward distinguishes them by leveraging the semantic difference between forward and reverse video plays. A response $\tilde{o}$ to the reversed video (e.g., describing "organizing or storing the box") is used as a negative reference. The reverse reward, $r_i^{\text{rev}} = 1 - \text{Similarity}(o_i, \tilde{o})$, then penalizes forward-video responses like $o_3$ if they incorrectly align with the reverse response $\tilde{o}$ (as indicated by red highlighting), thereby favoring temporally aware responses like $o_1$ that accurately reflect the forward video's AoT.

**Implementation**    As discussed in Sec. 3.2 of the main paper, $\text{Similarity}(\cdot, \cdot)$ return a similarity score between 0 and 1 and is implemented as follows. For MCQ tasks, it uses deterministic value checking (1.0 for a correct match, 0.0 otherwise); For open-ended QA and captioning, we employ Llama-3.1-70B-Instruct [24] as an LLM judge, which is prompted to output a semantic similarity score within the [0, 1] range, using the prompts below. The previously defined language query $l$, target response $o^\star$, candidate response $o_i$ and reverse response $\tilde{o}$ are referenced here.

---

**LLM-based Similarity Calculation**

**[Open-ended QA prompt]**
Please compare the following two answers for the question below and rate their similarity on a scale of 0 to 1.
Question: $l$
Answer 1: $o_i$
Answer 2: $\tilde{o}$
Output only a single numeric value between 0 and 1 (no additional text or explanation).

**[Captioning Prompt]**
Compare the following video caption with the ground truth caption and rate their similarity on a scale of 0 to 1.
Generated caption: $o_i$
Ground truth caption: $o^\star$
Output only a single numeric value between 0 and 1 (no additional text or explanation).

---

During training, candidate responses $o_i$ are generated with a temperature of 1.0 to encourage exploration. The reverse-conditioned response $\tilde{o}$ is generated deterministically (temperature set to 0). ArrowRL training for each model involves 2000 RL steps over approximately 3 days on 6 NVIDIA GH200 GPUs.

To ensure a fair comparison, inference settings for base LMMs and their ArrowRL-enhanced are identical, differing only by model checkpoint. Our default input configuration is 16 frames for LLAVA-OV-7B, and 1 FPS (with a maximum of 16 frames) for Qwen2-VL-7B and Qwen2.5-VL-7B. Benchmark-specific adjustments include processing up to 32 frames (sampled at 1 FPS for Qwen models) for TVBench (due to video length) and reporting Vinoground at 4FPS for Qwen models to align with [80] (further frame rate analysis in Fig. 11). For TempCompass, adhering to [82], we report only on its deterministic subtasks (multi-choice QA, yes/no QA, caption matching; 5536 samples).

### B.4   AoTBench

To evaluate AoT perception in LMMs, we construct AoTBench. The dataset composition is detailed in Table 6, with illustrative examples in Fig. 10. For the first and second T2V task, we concatenate the forward, a two-second black video and reversed video segments into a single input, and use the same prompt as Training Prompt I above, but with shuffled MCQ options to test model generalization.

Furthermore, Table 7 quantifies the increased AoT sensitivity of our selected VQA subset, showing significantly higher TDS values compared to the original benchmark.

## C   Additional Results

### C.1   Quantitative Results

**Further Results Analysis**    From Table 1 in the main paper, we see that ArrowRL appears to leverage inherent base model strengths for improvements. For instance, the visually adept Qwen2-VL-7B

**Table 6:** AoTBench Dataset Breakdown. The benchmark comprises three distinct tasks, derived from a diverse suite of video sources, designed to assess AoT awareness of LMMs.

| Task | Video Source | # VQA |
|------|--------------|-------|
| Sequence Direction Classification | ReverseFilm [66] | 144 |
| | UCF101 [58] | 500 |
| Directional Caption Matching (V2T) | RTime [18] | 1,992 |
| Directional Caption Matching (T2V) | RTime [18] | 1,992 |
| AoT-sensitive VQA | VITATECS [37], TemporalBench [9], NExT-QA [70], PerceptionTest [51], VideoMME [21], Vinoground [80], TempCompass [44], TVBench [15] | 1,800 |

🕒 **AoTBench**

**[Sequence Directionality Classification]** Two video segments are presented: one plays forward, the other in reverse. Which video segment is played in reverse?

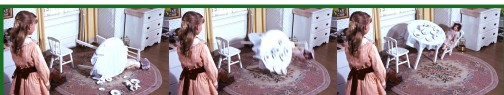 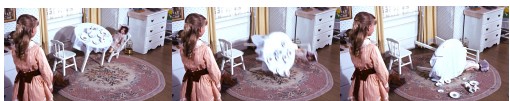

**[Directional Caption Matching – T2V]** Two video segments are presented: one plays forward, the other in reverse. Which video segment matches the caption "someone taking personal care items out of a clear plastic bag"?

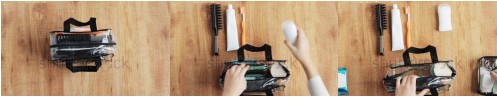 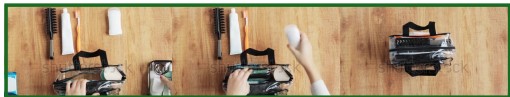

**[Directional Caption Matching – V2T]** Which caption best describes the video?
- A bulb is installed to the white round lamp shade by a person and is lighted up.
- A person removes the luminous bulb from the white round lamp shade with his hands.

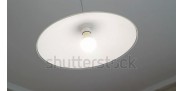 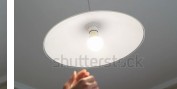 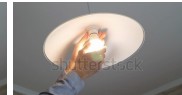 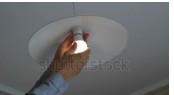 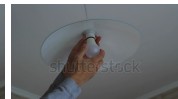 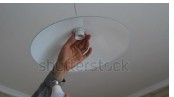

**[AoT-sensitive VQA]** What change is occurring to the 3D house model?
- being constructed
- being dismantled
- being renovated

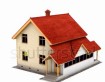 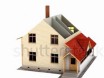 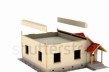 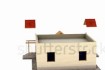 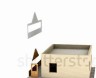 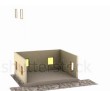

**Figure 10:** Visual overview of the three core task components in AoTBench, designed to evaluate different facets of AoT understanding in LMMs.

(evidenced by its base performance on T2V caption matching) sees a +21.0% gain with ArrowRL on the visually-focused sequence direction classification task. Meanwhile, Qwen2.5-VL-7B, which exhibits greater language proficiency (reflected in its base V2T caption matching scores), achieves its largest gains (+9.2%) on the more language-centric AoT-VQA task after ArrowRL enhancement.

**Frame Rate Analysis** Fig. 11 presents our inference frame analysis on Vinoground, comparing ArrowRL-enhanced Qwen2.5-VL-7B against its base model across different frame settings (1-4 FPS). Crucially, although ArrowRL training utilizes a fixed 16 frames per input (for efficiency), the performance gains provided by ArrowRL remain consistent across varying temporal granularities at inference, showcasing its generalization beyond the specific training setup, and giving evidence that the base models overlook temporal detail even with higher framerates.

**Table 7:** Comparing Temporal Divergence Score (TDS) averaged for all samples vs. top 200 selected high-TDS samples across nine existing VQA benchmarks. The selection process yields a 1,800-sample subset with substantially increased average TDS, specifically designed to challenge LMM temporal perception.

| | VITATECS | TemporalBench (S) | NExT-QA | PerceptionTest | VideoMME (S) | Vinoground (V) | Vinoground (T) | TempCompass | TVBench |
|---|---|---|---|---|---|---|---|---|---|
| All | 0.039 | 0.062 | 0.073 | 0.083 | 0.113 | 0.212 | 0.408 | 0.461 | 0.549 |
| Selected | 0.738 | 1.258 | 1.757 | 3.185 | 0.690 | 0.512 | 1.287 | 4.617 | 3.504 |

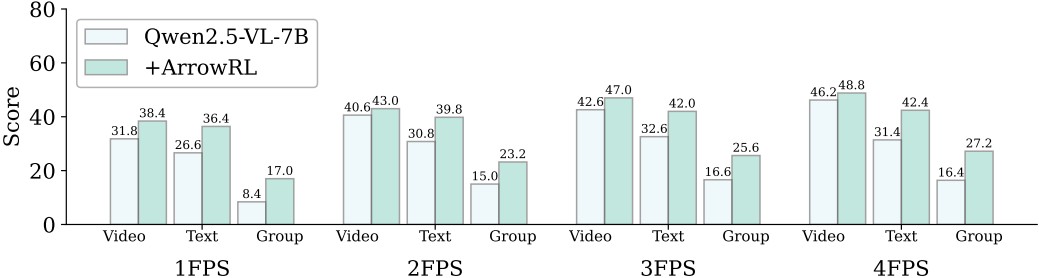

**Figure 11:** Impact of inference frame rate analysis on Vinoground. The ArrowRL's performance gains remain consistent across different frame settings (1-4FPS), showcasing its generalizability.

### C.2 Qualitative Results

**More Qualitative Results** Supplementing Fig. 8 in the main paper, Fig. 12 presents additional qualitative examples that reinforce the effectiveness of ArrowRL. These comparisons highlight how base LMMs often overlook temporal progression or direction, often relying on static cues or language biases. Conversely, our ArrowRL-enhanced models exhibit improved AoT sensitivity, leading to more accurate and temporally coherent responses across these challenging scenarios.

**Failure Cases** Fig. 13 presents one failure case on AoTBench, where neither the base LMM nor the ArrowRL-enhanced version answers correctly. The failure stems from the uniform sampling of 16 frames (6 visualized here) not capturing the key visual moments where a man in a dark suit holds a ring—a crucial cue for inferring he is about to get married. Such failures suggest potential avenues for future improvement, like advanced keyframe selection methods over uniform sampling, or the incorporation of auxiliary modalities such as audio, which in this case contains helpful cues.

## D Limitations

Our construction of AoTBench relies on using selected LMMs as evaluators, meaning its sensitivity is inherently dependent on the initial AoT perception capabilities of these models. Potentially challenging samples might be missed if current evaluator models universally fail to exhibit sensitivity (i.e., yield low TDS) despite underlying temporal relevance. Nevertheless, to provide a quantitative check of AoTBench, we manually verify a small subset and find our TDS-based selection of temporally challenging examples aligns with human judgment 44 out of 50 times, suggesting reasonable concordance where models do possess baseline sensitivity.

Additionally, the reward calculation for ArrowRL utilizes an auxiliary LLM to generate similarity scores; this dependence on an LLM judge, while common [86, 25, 71], can introduce a degree of

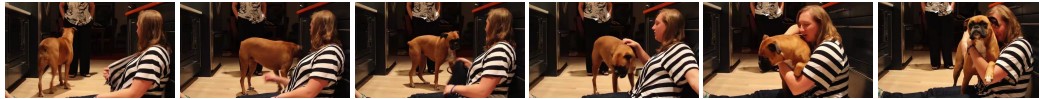

**Q.** Why does the dog walk toward the lady after the lady reaches her hand toward it?

**Qwen2-VL-7B:** to catch the food

**Qwen2-VL-7B + ArrowRL:** let the lady hug it

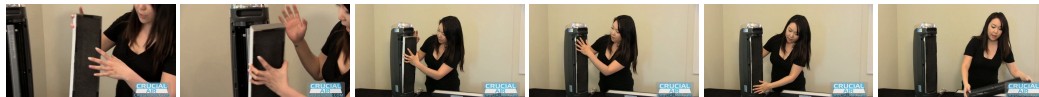

**Q.** Which caption best describes this video?

**Qwen2.5-VL-7B:** A woman is speaking while she is handling an air filter with both hands. Using her right hand to hold in place the filter device, the left hand pulls the filter out of its place in the device. Both hands then pick up the outer shield of the filter from the table.

**Qwen2.5-VL-7B + ArrowRL:** A woman is speaking while she is handling an an air filter with both hands. Using her right hand to hold in place the filter device, the left hand pushes the filter into its place in the device. Both hands then pick up the outer shield of the filter from the table.

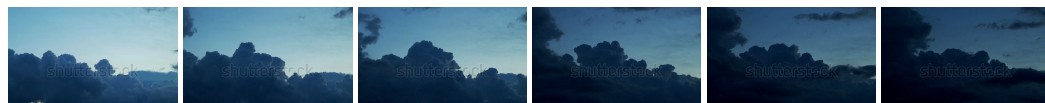

**Q.** Provide a one-sentence description of the video, focusing solely on the primary actions taking place.

**LLaVA-OV-7B:** The video captures the serene and dynamic movement of clouds against a tranquil blue sky, creating a peaceful and picturesque scene.

**LLaVA-OV-7B + ArrowRL:** The video features a series of images capturing the movement of clouds in the sky, with the clouds transitioning from lighter to darker shades as the light diminishes.

**Figure 12:** Additional qualitative examples comparing base LMMs with their ArrowRL-enhanced counterparts. ArrowRL enables models to succeed on AoT-sensitive VQA and produce temporally coherent captions, while base LMMs often struggle with understanding temporal progression.

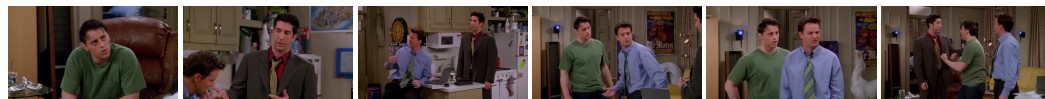

**Q.** According to the video, who is about to get married?

**A.** The man in the dark suit.

**Qwen2.5-VL-7B:** It is unclear

**Qwen2.5-VL-7B + ArrowRL (ours):** It is unclear.

**Figure 13:** A failure case from AoTBench. Uniformly sampling 16 frames (6 visualized) fails to capture the critical visual moments (i.e., the man in the dark suit holding a wedding ring).

uncertainty or potential bias into the reward signal. Nevertheless, we found this approach viable for our purposes, as the LLM performs a relatively easy, straightforward text-to-text similarity assessment. We empirically find that (also illustrated in Fig. 6 of the main paper and Fig. 9) the resulting similarity scores, which underpin our reward calculation, are consistent and appear reasonable.

Furthermore, our current investigation focuses on short videos, a scope chosen due to the significant temporal challenges already evident in these scenarios for leading LMMs. We view this work as a initial step towards improving AoT understanding. Future directions include extending these methods to long videos (potentially incorporating keyframe selection) and exploring integration with temporal reasoning tasks that necessitate explicit reasoning traces (e.g. Chain-of-Thought).

# E   Societal Impacts

This work focuses on improving a fundamental aspect of temporal perception (AoT sensitivity) in LMMs. Positive impacts stem from creating more reliable and rational AI systems. Better AoT perception can lead to LMMs with more accurate internal world models, improving their utility in tasks requiring understanding of processes, procedures, or event timelines. This could benefit applications like education, assistive technology and robotics. The primary risks are associated with the general capabilities of advanced LMMs rather than AoT sensitivity specifically. Any improvement could potentially be misused if integrated into systems for generating disinformation or invasive surveillance, though our method doesn't directly enable these.

