# OpenReview forum: "Seeing the Arrow of Time in Large Multimodal Models"
_NeurIPS.cc/2025/Conference — NeurIPS 2025 poster_

### Official Review · Reviewer_GqEN · 2025-06-28

**Clarity:** 4
**Significance:** 3
**Originality:** 3
**Rating:** 5
**Confidence:** 5

**Summary:**

**Problem**: In this work, the authors highlight the problem of modern large multimodal models (LMMs) not using the "arrow of time" while perceiving videos.

**Key contributions**: First, it is shown that existing LMM benchmarks (e.g., TemporalBench) and LMM models (e.g., Qwen2-VL) are not sufficiently sensitive to the arrow of time, i.e., shuffling or reversing the frame order does not substantially affect the accuracy. Second, as a redressal, an RL based post-training strategy is proposed to instil this time-sensitivity into an existing LMM (e.g., Qwen2-VL). Two reward signals are used: target fidelity (to encourage the correct answer) and reverse reward (to force different answers if the video is presented in a reverse order). Third, a new benchmark, AoTBench is proposed with three sub-tasks: (i) direction classification: whether a video if played forward/reverse, (ii) caption matching: matching forward/reversed videos to the right language description, (iii) VQA samples from existing benchmarks that are more likely to be sensitive to the arrow of time.

**Main outcome:** It is experimentally shown that the proposed strategy not only improves performance on the proposed AoTBench, but also improves performance on existing video benchmarks like TV-Bench and Vinoground.

**Questions:**

**Questions**

Some of these are already mentioned in the weaknesses section. I have elaborated a bit on them here.

1. Why should an existing video LLM model understand what one means by "forward or reverse direction of the video"? I do agree that checking understanding of arrow of time for modern language-based video models is essential.
2. Describing direction of time in language is tricky for irreversible videos. How would one describe a reversed video of "smoke diffusing"? Although I know this is not the main task, but it is worth asking how the model's perception changes when encountering a reversed video (e.g., are certain activations activated more when this happens?)
3. Does the added post-training affect "spatial" performance at all? For example, on non temporal benchmarks, does the model suffer at all or retain its performance?

**Suggestions**

1. Reversed videos may be out of distribution for existing models. Perhaps, is there a control task (like in [1]) where the question is insensitive to the arrow of time (e.g., describe all objects appearing in the video) and the model does well with both forward and reverse videos? This can establish that low performance of existing models need not be because the video is OOD.
2. A justification for the choice of measure (KLD between output distributions of **first token**) of arrow of time consistency would make the paper stronger. Likewise, it would be helpful to clarify L145 on filtering out samples.

[1] Test of Time: Instilling Video-Language Models with a Sense of Time. Bagad et al. CVPR 2023.

**Ethical Concerns:**

["NO or VERY MINOR ethics concerns only"]

**Final Justification:**

I have retained my original positive rating of the paper.

From the initial review, all my concerns were addressed by the rebuttal. Particularly, we had a fruitful discussion on the aspect of grounding physics (arrow of time in particular) in language.

I also went through all the other reviews and the corresponding rebuttal responses. Reviewer `mxwr` raises a fair question on whether encouraging  dissimilarity between forward and backward responses is essential. In my view, the fact that the proposed RL post-training improves arrow of time reasoning while maintaining performance on standard benchmarks is a major positive. One can see arrow of time as a fundamental desirable property which may not be applicable to everyday videos, but can be fundamental to understanding intuitive physics, for example. So instilling arrow of time while doing well on standard benchmarks is a good outcome in my view. Overall, I think the authors addressed all questions quite comprehensively.

**Limitations:**

While the Supp includes failure cases, it may be worth adding a couple of lines in the main text on what kinds of samples the model fails on and if there is a pattern to it. That could also guide future work.

**Quality:**

3

**Strengths And Weaknesses:**

**Strengths**
1. Elegant adaptation of GRPO for time-sensitive video understanding. While the problem of arrow of time is not new, I think this paper presents a thorough method to tackle it in context of modern video-LLMs. The "reverse reward" for distinctive answers for forward/reverse videos is a sensible choice.
2. The weaknesses shown in existing video models/benchmarks should guide the video community towards better models/benchmarks. For example, the insight that LLAVA-OV-7B, Qwen2.5-VL-7B are more time-sensitive models (on TVBench) and Vinoground/TVBench/TempCompass are more time-sensitive benchmarks is very useful.
3. The design of new AoTBench and the subsequent experiments are very thorough and well thought-out. The proposed ArrowRL shows clear benefits on arrow of time tasks while also improving performance on existing benchmarks.
4. The paper is written very well; the flow is very smooth and logical.


**Weaknesses**
1. Grounding arrow of time for irreversible videos in language is probably ill-posed. In the first AoTBench task, the authors propose a VQA-like task where the model is tasked with choosing which of the two videos is played in forward/reverse directions.
	1. For irreversible videos (e.g., where reverse video violates a physical law), the task is purely visual. It is awkward to describe a waterfall going upwards in language.
	2. Why should an existing video LLM model understand what one means by "forward or reverse direction of the video"? In this work, the train set contains such questions. So the resulting model knows what "forward" means. But what about existing models which may not have "forward"/"reverse" in their train set?
	3. Reversed videos may be out of distribution for existing models. Could the low performance be attributed to that instead of just not understanding arrow of time?
2. The metric designed to measure if the LMM understands arrow of time is slightly unconvincing. As described in L140-L145, the probability of the first token over the MCQ options is considered. A justification for this is lacking. Also, "We first filter out instances where the average reverse performance exceeds its forward performance." -- this is confusing.

---

> ### Author Rebuttal · Authors · 2025-07-30
>
> We thank reviewer GqEN for the helpful comments and for providing thoughtful feedback on our work.
>
> ---
>
> **1. On Grounding Arrow of Time in Language**
>
> > Why should an existing video LLM model understand what one means by "forward or reverse direction of the video"?
>
> > Describing direction of time in language is tricky for irreversible videos. How would one describe a reversed video of "smoke diffusing"?
>
> We thank the reviewer for these insightful comments. They touch on the core challenge of evaluating a concept as fundamental as the Arrow of Time within a language-based framework. We address the key points below.
>
> **Describing Arrow of Time in Language**. We agree that describing a physically irreversible event in reverse (like a waterfall going up) is linguistically awkward. Our evaluation is designed accordingly to ensure that it never requires a model to generate a free-form description of a physically implausible reversed video:
> - The quantitative sequence direction classification task in AoTBench is a classification, not a captioning task. LMMs are required to map its visual understanding of sequence directionality to a simple linguistic label ("forward" or "reverse"), testing a fundamental aspect of multimodal reasoning.
> - For qualitative examples that do involve description (e.g., *Fig. 1, Fig. 7, right*), we use "reversible" videos where both directions are physically plausible. We will add clarification in the paper.
>
> To your specific question about describing reversed physically implausible events such as "smoke diffusing", our observation is that current LMMs, due to lack of temporal directionality understanding, fail to describe the implausible dynamics. They tend to focus on static objects and often repeat the forward-action verb (e.g., still saying "smoke diffusing"), confirming their temporal insensitivity.
>
> In all, we believe the ability to recognize that a video violates the physical laws of nature and connect that visual intuition to a symbolic linguistic concept is a fundamental capability. For LMMs to progress, they need to be able to ground simple linguistic concepts like "forward" and "reverse" in their visual understanding—a task humans perform effortlessly. Our classification task provides a crucial testbed for this capability.
>
> **On Evaluating Base LMMs**. About whether an existing LMM should be expected to understand this task without explicit training, we clarify that our goal in testing base LMMs is not to claim they should succeed, but rather to use the task as a diagnostic tool. Their near-random performance, as shown in *Table 1*, is a key finding of our paper. It provides a critical baseline that demonstrates a clear capability gap in current models and powerfully motivates the need for targeted training methods like ArrowRL.
>
> Finally, we agree with the reviewer's insightful point that understanding when a visual phenomenon is difficult to describe is a key challenge for more advanced LMMs. While our work focuses on the foundational skill of recognizing directionality, we believe exploring the limits of linguistic description for complex physical events is an important direction for future research. We welcome any further suggestions from the reviewer on improving this task format and will revise the paper accordingly to reflect this discussion.
>
> ---
>
> **2. Control Task to Disentangle OOD Effects**
>
> > Is there a control task (like in [1]) where the question is insensitive to the arrow of time (e.g., describe all objects appearing in the video) and the model does well with both forward and reverse videos? This can establish that low performance of existing models need not be because the video is OOD.
>
> We appreciate the suggestion of adding a control task, which is well-motivated by prior work ([1] Bagad et al., cited in our paper as [2]) on disentangling temporal effects. Accordingly, we design action recognition as the control task, using videos from the UCF101 dataset and their action label. For each video, we formulate a binary-choice question with the prompt, "What action is being performed in this video?" The correct action was provided as one choice, and the incorrect option was randomly drawn from the set of all other action labels. We then evaluated several LMMs on both the original (forward) and time-reversed versions of these videos.
>
> | LMM               | Forward Acc. (%) | Reverse Acc. (%) |
> |-------------------|:------------------:|:-------------------:|
> | GPT-4o            | 100.0            | 100.0             |
> | Gemini-1.5-Pro    | 98.2             | 99.4              |
> | LLaVA-OV-7B       | 99.2             | 99.4              |
> | Qwen2.5-VL-7B     | 99.4             | 99.4              |
>
> The results above show that all models perform at near-perfect accuracy on this control task, irrespective of video direction. This contrasts sharply with their near-random performance on our AoT-sensitive sequence direction classification task (*Table 1*, UCF column). This analysis provides strong evidence that the failure of existing LMMs on AoT tasks is due to their inability to perceive temporal directionality, not a general failure to process reversed videos as OOD inputs. We appreciate the reviewer for prompting this important analysis and will incorporate it into our paper.
>
> [1] Test of Time: Instilling Video-Language Models with a Sense of Time. Bagad et al. CVPR 2023.
>
> ---
>
> **3. Justification for Using the First Token Probability and Clarification on Ln 145**
>
> > A justification for the choice of measure (KLD between output distributions of first token) of arrow of time consistency would make the paper stronger. Likewise, it would be helpful to clarify L145 on filtering out samples.
>
> **First Token Probability**. We use the first-token probability to estimate the model’s likelihood of selecting each option in the MCQ format. Since our prompt instructs the model to “Answer with the option’s letter from the given choices directly,” the probability of the first token (e.g., A, B) serves as a reliable proxy for the model’s chosen answer. This approach is standard and widely adopted by the LLM community [1–3]. Our novelty lies in adapting this standard evaluation practice for temporal analysis by computing the KL Divergence between the distributions generated from the forward vs. the reversed video.
>
> **Ln145 Clarification**. In some rare cases, we observe that our candidate LMMs perform better on reversed videos than on the original forward ones. These are typically outlier cases where temporal direction is either ambiguous or irrelevant. We exclude these samples from further analysis to ensure that our evaluation focuses on instances where AoT reasoning is actually required.
> We appreciate the reviewer’s suggestion and will update the manuscript to clarify these design choices.
>
> [1] Brown et al., Language Models are Few-Shot Learners, NeurIPS 20
>
> [2] Hendrycks et al., Massive Multitask Language Understanding, ICLR 21
>
> [3] Lin et al., TruthfulQA: Measuring How Models Mimic Human Falsehoods, ACL 22
>
> ---
>
> **4. General Video Benchmark Performance**
>
> > On non temporal benchmarks, does the model suffer at all or retain its performance?
>
> We thank the reviewer for this important question regarding ArrowRL’s performance beyond temporally-focused benchmarks. We reiterate that ArrowRL is designed as a targeted enhancement for an LMM's temporal sensitivity. As our analysis in *Sec. 3.1* shows, many existing video benchmarks do not actually require this capability to be solved (e.g., they can be answered with single, shuffled or reversed frames). For this reason, our main results in *Table 1* focus on the benchmarks we identified as temporally sensitive, where ArrowRL shows significant boosts.
>
> The reviewer raises a valuable follow-up question: whether our specialized training harms performance on easier tasks where AoT is not required. To answer this, we have conducted experiments on the benchmarks that our own analysis identified as less sensitive (*Fig. 3 right*).
>
> | Benchmark         | Base LMM | w/ ArrowRL |
> |-------------------|:------------------------:|:------------:|
> | VideoMME (S)      | 67.78                  | 68.11      |
> | NExT-QA           | 77.11                  | 78.11      |
> | TemporalBench (S) | 60.57                  | 61.17      |
> | VITATECS          | 84.66                  | 85.37      |
> | PerceptionTest    | 57.48                  | 57.64      |
>
> As shown, ArrowRL preserves or even slightly improves performance across these general video benchmarks that do not demand temporal awareness. This new analysis (same as shown for Reviewer mxwr and 6fDT) confirms that ArrowRL is a targeted enhancement: it provides great gains on temporally-aware benchmarks (as shown in *Table 1* with data from 3 relevant and popular benchmarks) without degrading performance on more general benchmarks where AoT is not a primary factor (*Ln 280-283*). We thank the reviewer for this suggestion and will add this analysis to the final paper.
>
> ---
>
> **4. On Including Analysis of Failure Cases in Main Text**
>
> > While the Supp includes failure cases, it may be worth adding a couple of lines in the main text on what kinds of samples the model fails on and if there is a pattern to it. That could also guide future work.
>
> Thank you for your comment. We will move a concise summary of our failure case analysis (detailed in *Supp. Sec. D*) into main paper as suggested.

---

> > ### Comment · Reviewer_GqEN · 2025-08-01
> > **Acknowledgment of the rebuttal**
> >
> > I thank the authors for providing a detailed and comprehensive rebuttal.
> >
> > **1. Grounding time in language**
> >
> > > we believe exploring the limits of linguistic description for complex physical events is an important direction for future research
> >
> > I think this summarises my broader point quite well. Perhaps, physical events could be best described by equations of physical laws and not English language. In case of arrow of time, it may be an interesting future direction to predict what physical law has been violated and why if a video is reversed.
> >
> > On the bit about evaluating base LMMs, (and also related to (3.) about using first token probability), I understand the rational better now. I would add that another interesting direction could be to monitor if certain specialised neurons are consistently activated in an LMM upon processing a time-reversed (physically implausible video) similar to this study in humans [1].
> >
> > Overall, I am satisfied by the answers provided by the authors.
> >
> > [1] The arrow of time of brain signals in cognition: Potential intriguing role of parts of the default mode network. Gustavo Deco, Yonatan Sanz Perl, Laura de la Fuente, Jacobo D Sitt, B T Thomas Yeo, Enzo Tagliazucchi, Morten L Kringelbach. Network Neuroscience. 2023.
> >
> >
> > **2. Control Task to Disentangle OOD Effects**
> >
> > I am satisfied by the proposed contol experiment.
> >
> > **3. Justification for Using the First Token Probability and Clarification on Ln 145**
> >
> > This is a fair justification.
> >
> > **4. General Video Benchmark Performance**
> >
> > It is a strong result that the model does not deteriorate on benchmarks that are heavier on spatial understanding.

---

> > > ### Author Response · Authors · 2025-08-01
> > >
> > > We are very grateful for the reviewer's positive and thoughtful engagement with our work, and we are pleased that our rebuttal was convincing.
> > >
> > > We appreciate the great suggestions for future work. The ideas of predicting which physical laws are violated, and taking inspiration from neuroscience to find specialized neurons, are fascinating. They provide a wonderful perspective on future research, and we will be sure to incorporate this discussion into our paper's conclusion.
> > >
> > > Thank you again for your time and for helping us strengthen our paper.

---

### Official Review · Reviewer_6fDT · 2025-07-02

**Clarity:** 3
**Significance:** 2
**Originality:** 3
**Rating:** 4
**Confidence:** 5

**Summary:**

This paper identifies the problem that existing Video LLMs are insufficiently sensitive to the temporal directionality of events in videos. It designs ArrowRL to enhance the perception of temporal directionality in Video LLMs through a reverse reward mechanism, and develops AoTBench to evaluate this capability. Experiments show that the method significantly improves model performance on time-related tasks.

**Questions:**

* How did the authors construct response pairs where the difference between reverse responses and forward responses can reflect temporal directionality?
* Does this method still show improvements on general temporal benchmarks like VideoMME?

**Ethical Concerns:**

["NO or VERY MINOR ethics concerns only"]

**Final Justification:**

The authors have comprehensively addressed my questions:

1. Regarding the question of whether "the difference between reverse responses and forward responses can reflect temporal directionality," the authors have resolved it through "video source selection" and "dynamically weighting forward/reverse rewards." Overall, this is reasonable.
2. The training scheme introduced by the authors can slightly improve the performance on General Video Benchmark.

Therefore, I retain my score as positive.

**Limitations:**

Yes

**Quality:**

3

**Strengths And Weaknesses:**

### Strengths

* The problem identified by the authors is mentioned in previous Video Benchmarks (e.g., Tempcompass), but no related work has attempted to solve it. This paper fills this gap, demonstrating significant research significance.
* The authors' method is rationally designed: leveraging the intuition that "forward-time sampled outputs should differ substantially from reverse-time ground truth outputs", they use an LLM to judge the similarity between the two as a reward to supervise Video LLMs in enhancing their temporal direction perception.
* The authors provide AoTBench to systematically quantify Video LLMs' understanding of temporal order, effectively drawing the community's attention to this issue.
* The experimental results are solid, showing notable improvements for SOTA models on temporal benchmarks (e.g., Qwen2.5VL achieves a 1.7% improvement on Tempcompass).

### Weaknesses

* Whether the difference between Reverse Response and Forward Response can reflect temporal directionality is crucial to the method's effectiveness, yet the authors lack description on how to construct reverse-forward response pairs that meet this requirement.
* The method proposed in the paper mainly improves benchmark performance for tasks explicitly related to temporal directionality (e.g., temporal direction classification, forward-reverse temporal caption matching), but it remains unclear whether it can enhance the general ability to understand temporal information in videos.

---

> ### Author Rebuttal · Authors · 2025-07-30
>
> We thank reviewer 6fDT for the helpful comments and for providing thoughtful feedback on our work.
>
> ---
>
> **1. Clarification Regarding ArrowRL Response Generation**
>
> > How did the authors construct response pairs where the difference between reverse responses and forward responses can reflect temporal directionality?
>
> We appreciate this insightful question, and fully agree that ensuring the difference between forward and reverse responses reflects true temporal directionality is crucial.
>
> First, we clarify that our method does not construct or use pre-defined "forward-reverse response pairs." Instead, as part of the GRPO framework, all responses are generated on-the-fly during training: the policy model generates a group of forward responses to the original video, and a reverse response. These responses are then used to compute the fidelity and reverse rewards in real time.
>
> This dynamic approach means the central challenge is ensuring the reverse reward is applied intelligently, which is crucial for the method's effectiveness. We achieve this with two techniques:
>
> **High-temporality Data Curation**. For captioning tasks, we leverage the curated RTime dataset [14], which consists of videos specifically selected for having meaningful and distinct forward vs. reverse actions (*Ln 251-253*). This provides a clean, high-quality signal for learning.
>
> **Dynamic Reward Weighting**. For larger, less curated datasets, we employ a dynamic weighting scheme (*Ln 207–214*) that detects when the reversed response is too similar to the forward ground truth and automatically disables the reverse reward for that sample.
> The effectiveness of these strategies is supported by our experiments.  The ablation in *Table 2* (+ArrowRL (γ=1)) already demonstrates that removing the dynamic weighting hurts performance. Furthermore, to isolate the effect of data selection, we ran a new experiment focused only on the captioning task, replacing the high-quality RTime data with uncurated videos from LLaVA-Video-178K. The results below (same as shown for Reviewer tJ9G) underscore the importance of our data curation strategy.
>
> | Model                          | AoTBench Acc. (%) |
> |-----------------------------|:-----------------:|
> | Base LMM                      |       56.2        |
> | w/ ArrowRL (RTime Cap.)            |       **60.4**        |
> | w/ ArrowRL (LLaVA-Video-178K Cap.) |       57.7        |
>
> Together, these strategies ensure our reverse reward is applied precisely and only to cases where it can encourage a meaningful understanding of temporal directionality.
>
> ---
>
> **2. General Video Benchmark Performance**
>
> > Does this method still show improvements on general temporal benchmarks like VideoMME?
>
> We thank the reviewer for this important question regarding ArrowRL’s performance beyond temporally-focused benchmarks. We reiterate that ArrowRL is designed as a targeted enhancement for an LMM's temporal sensitivity. As our analysis in *Sec. 3.1* shows, many existing video benchmarks do not actually require this capability to be solved (e.g., they can be answered with single, shuffled or reversed frames). For this reason, our main results in *Table 1* focus on the benchmarks we identified as temporally sensitive, where ArrowRL shows significant boosts.
>
> The reviewer raises a valuable follow-up question: whether our specialized training harms performance on easier tasks where AoT is not required. To answer this, we have conducted experiments on the benchmarks that our own analysis identified as less sensitive (*Fig. 3 right*), including VideoMME as suggested by the reviewer.
>
> | Benchmark         | Base LMM | w/ ArrowRL |
> |-------------------|:------------------------:|:------------:|
> | VideoMME (S)      | 67.78                  | 68.11      |
> | NExT-QA           | 77.11                  | 78.11      |
> | TemporalBench (S) | 60.57                  | 61.17      |
> | VITATECS          | 84.66                  | 85.37      |
> | PerceptionTest    | 57.48                  | 57.64      |
>
> As shown, ArrowRL preserves or even slightly improves performance across these general video benchmarks that do not demand temporal awareness. This new analysis (same as shown for Reviewer mxwr) confirms that ArrowRL is a targeted enhancement: it provides great gains on temporally-aware benchmarks (as shown in *Table 1* with data from 3 relevant and popular benchmarks) without degrading performance on more general benchmarks where AoT is not a primary factor (*Ln 280-283*). We thank the reviewer for this suggestion and will add this analysis to the final paper.

---

> ### Comment · Reviewer_6fDT · 2025-08-05
>
> The authors have comprehensively addressed my questions:
>
> * Regarding the question of whether "the difference between reverse responses and forward responses can reflect temporal directionality," the authors have resolved it through "video source selection" and "dynamically weighting forward/reverse rewards." Overall, this is reasonable.
> * The training scheme introduced by the authors can slightly improve the performance on General Video Benchmark.
>
> Therefore, I retain my score as positive.

---

> > ### Author Response · Authors · 2025-08-06
> >
> > We are very grateful for the reviewer's positive and thoughtful engagement with our work. We are pleased that our clarifications regarding the reward mechanism and the experimental results on general video benchmarks were clear and convincing.
> >
> > Thank you again for your time and for helping us strengthen our paper.

---

### Official Review · Reviewer_mxwr · 2025-07-03

**Clarity:** 2
**Significance:** 2
**Originality:** 3
**Rating:** 3
**Confidence:** 4

**Summary:**

This paper addresses the temporal directionality in LMMs for video understanding. The authors show that existing LMMs often fail to distinguish between forward and reversed videos, leading to semantic errors in tasks such as question answering or captioning. To tackle this, they propose ArrowRL, a reinforcement learning approach using a novel reverse reward that encourages models to produce different responses for forward versus reversed inputs. They also introduce AoTBench, a benchmark specifically designed to evaluate AoT sensitivity with tasks like sequence direction classification, directional caption matching, and AoT-sensitive VQA. Extensive experiments show that ArrowRL significantly improves AoT perception across multiple open-source LMMs and outperforms baselines on both AoTBench and existing VQA datasets.

**Questions:**

Why, in Fig. 3 (left), are the forward and reverse results similar? How should this be explained?

**Ethical Concerns:**

["NO or VERY MINOR ethics concerns only"]

**Final Justification:**

After reading the feedbacks from authors, I was convinced that the proposed technique to enhance the model’s reverse-time understanding capability is helpful. However, the overall effectiveness is marginal. I would like to increase the score to 3.

**Limitations:**

yes

**Quality:**

3

**Strengths And Weaknesses:**

Strengths:

(1) The paper demonstrates that AoT sensitivity is a fundamental yet neglected requirement for video LMMs.

(2) ArrowRL introduces a novel reverse reward in a GRPO framework to explicitly teach temporal directionality, which is an original and well-motivated idea.

(3) The authors show consistent improvements across three base LMMs, multiple AoTBench tasks, and existing video QA benchmarks, demonstrating strong generalization.

(4) The paper contributes a new, carefully curated benchmark with multiple sub-tasks targeting AoT sensitivity, filling a clear evaluation gap in the field.

Weaknesses:

(1) Most of the multiple-choice questions do not require temporal reasoning and therefore do not clearly demonstrate the need for time awareness. As a result, it is not convincing to use MCP as examples to illustrate the motivation for AoT.

(2) The formulation is confusing, e.g., what does P denote in line 167?

(3) Applying GRPO (or other RL methods) may not be necessary. Augmenting the pre-training (or SFT) datasets with reversed videos could potentially address this problem.

(4) Why is maximizing the dissimilarity between forward and backward responses essential? Enforcing them to be dissimilar does not guarantee that the reversed response will be correct.

(5) The evaluation is not comprehensive. It lacks comparison with other popular video benchmarks (e.g., MVBench, VideoMME).

---

> ### Author Rebuttal · Authors · 2025-07-30
>
> We thank reviewer mxwr for the helpful comments and for providing thoughtful feedback on our work.
>
> ---
>
> **1. On the Use of MCQ for Evaluating Temporal Awareness**
>
> > Most of the multiple-choice questions do not require temporal reasoning and therefore do not clearly demonstrate the need for time awareness.
>
> Thank you for highlighting this critical point, which in fact perfectly states the central motivation for our work. We agree entirely that most existing video MCQ samples do not adequately test for temporal awareness. This widespread issue is exactly what inspired us to develop a more rigorous approach. In *Sec. 3.1 & 3.3*, we detail our diagnostic analysis of existing benchmarks, propose the Temporal Divergence Score (TDS) metric, which sifts through these inadequate benchmarks and algorithmically identifies the set of questions that are truly sensitive to the Arrow of Time.
>
> For the evaluation itself, as noted in *Supp. Ln 344-346*, we chose the MCQ format because it is the standard for objective and reproducible LMM evaluation. In contrast, open-ended QA and captioning are notoriously difficult to evaluate and often require LLM-based or human judges, which can introduce noise and reduce comparability [12].
>
> Therefore, the combination of our targeted TDS filtering with a standard, objective evaluation format provides what we are confident is a robust and convincing test for the Arrow of Time. If there are particular aspects about our evaluation that the reviewer finds unconvincing, we would be glad to elaborate further.
>
> ---
>
> **2. Clarification on Notation $P$**
>
> > what does P denote in line 167?
>
> We follow the convention used in the original GRPO paper [47], where $P$ is used to denote the underlying data distribution of a dataset. In our context, $P$ refers to the distribution from which we draw our training samples, each consisting of a question $q$ and response $o$ (*Ln 165-166*). We will revise the sentence to improve clarity.
>
> ---
>
> **3. On the Necessity of GRPO**
>
> > Applying GRPO (or other RL methods) may not be necessary. Augmenting the pre-training (or SFT) datasets with reversed videos could potentially address this problem.
>
> We appreciate the comment and agree that SFT is a critical comparison point. In fact, we included this exact experiment in our paper to test the necessity of our RL-based approach.
> As shown in *Table 2*, we trained a model using SFT on the exact same instruction tuning datasets designed for ArrowRL. While this approach improves upon the base LMM (57.4% vs. 56.2%), confirming the value of our training data, ArrowRL demonstrates a significant advantage, achieving 60.7%. This result indicates that RL is necessary to unlock the full potential of the training signal.
>
> ---
>
> **4. Clarification on ArrowRL reward**
>
> > Why is maximizing the dissimilarity between forward and backward responses essential? Enforcing them to be dissimilar does not guarantee that the reversed response will be correct.
>
> We fully agree with the reviewer that maximizing the dissimilarity between forward and backward responses would not guarantee correctness. Our methodology was designed with this principle in mind. As detailed in *Sec. 3.2*, ArrowRL’s reward is composed of two parts: (1) a fidelity reward ensures the forward response is correct by rewarding similarity to the ground truth, (2) a reverse reward encourages this correct response to be temporally sensitive by rewarding dissimilarity from reverse response. It is important to clarify that our objective is to generate correct and temporally-aware responses for forward videos, not to ensure the correctness of responses for reversed videos. The reversed video is used solely as a training signal (*Ln 202-214*).
>
> The final reward is a weighted sum of both components. *Fig. 4* and *Supp. Fig. 8* illustrate these two reward signals, and *Table 2* (along with our response #3 to Reviewer tJ9G) analyzes the effect of varying the weight between them. We thank the reviewer again and are happy to discuss further if there are additional concerns.
>
> ---
>
> **5. Clarification on Evaluation**
> > The evaluation is not comprehensive. It lacks comparison with other popular video benchmarks (e.g., MVBench, VideoMME).
>
> We appreciate this comment and would like to clarify our principled approach to benchmark selection, which draws from 8 popular existing benchmarks, and then present additional results as requested.
>
> **Principled Benchmark Selection**. As noted in *Ln 46-54*, *Fig. 2* and *Sec. 3.1*, a core finding of our work is that many existing video benchmarks are not well-suited to evaluate temporal sensitivity. To address this, we introduced the Temporal Divergence Score (TDS) to quantify each benchmark's reliance on the Arrow of Time. Our analysis of 8 existing benchmarks (*Fig. 3 right*) revealed that TVBench, TempCompass, and Vinoground are the most temporally sensitive benchmarks. We notably selected TVBench over the older MVBench, as TVBench was specifically designed to address the temporal insensitivity of its predecessor while using much of the same underlying video data. We therefore focused our main evaluation (*Table 1*) on these benchmarks to provide the clearest measure of improvement on temporal awareness (*Ln 157-161, 264-265*).
>
> **Additional Evaluation on General Benchmarks**. To address the reviewer's question about performance on other popular benchmarks that don’t require temporal reasoning (like VideoMME), we have conducted the requested experiments. The question is whether our method harms performance on tasks that do not require AoT reasoning. The results below show that it does not; in fact, we see neutral to slightly positive results across the board.
>
>
> | Benchmark         | Base LMM | w/ ArrowRL |
> |-------------------|:------------------------:|:------------:|
> | VideoMME (S)      | 67.78                  | 68.11      |
> | NExT-QA           | 77.11                  | 78.11      |
> | TemporalBench (S) | 60.57                  | 61.17      |
> | VITATECS          | 84.66                  | 85.37      |
> | PerceptionTest    | 57.48                  | 57.64      |
>
> These results demonstrate that ArrowRL is a targeted enhancement: it provides great gains on temporally-aware benchmarks (as shown in *Table 1* with data from 3 relevant and popular benchmarks) without degrading performance on more general benchmarks where AoT is not a primary factor (*Ln 280-283*). We thank the reviewer for this suggestion and will add this analysis to the final paper.
>
> ---
>
> **6. Clarification on Fig. 3**
>
> > Why, in Fig. 3 (left), are the forward and reverse results similar? How should this be explained?
>
> Thank you for your question. Your observation that the forward and reverse results are similar for many of the models shown is entirely correct, and this is precisely the finding that the figure is designed to illustrate.
>
> Our *Fig. 3* provides the temporal sensitivity analysis. As noted in *Ln 128-136, 155-161*, it shows that most existing LMMs are largely insensitive to temporal direction (left) and that current benchmarks fail to meaningfully evaluate AoT perception (right). In other words, even when provided with semantically distinct forward and reverse videos, these models often generate nearly identical responses—demonstrating a lack of temporal awareness, likely caused by an over-reliance on language priors, where the model's textual knowledge overrides the visual evidence of temporal direction (*Fig. 1(b), Ln 46–64*).
>
> This observed limitation directly motivates our work: we propose ArrowRL to improve temporal sensitivity in LMMs, and AoTBench as a focused benchmark to evaluate this capability. We thank the reviewer again and are happy to discuss further if there are additional concerns.

---

> > ### Comment · Reviewer_mxwr · 2025-08-05
> >
> > Thank you for the authors’ response. I agree that the proposed technique to enhance the model’s reverse-time understanding capability is helpful, but its impact seems marginal. I still have doubts about its overall effectiveness, as the forward pass alone should suffice for comprehensive video understanding. I will increase my score to 3.

---

> > > ### Author Response · Authors · 2025-08-06
> > >
> > > We are pleased that the reviewer found our rebuttal comprehensive and has raised their score; we appreciate their willingness to re-evaluate our work.
> > >
> > > Regarding the remaining concern about the impact being "marginal" and the sufficiency of the "forward pass alone," our ablation studies (Table 2 in the paper) were designed to test this exact hypothesis. The experiment where we remove the reverse reward entirely (α = 0 row) serves as a direct test of a "forward-only" RL approach.
> > >
> > > The results show this "forward-only" model performs much worse than our full method (with the reverse reward), with an accuracy of 55.6% compared to 60.7%. This provides direct evidence that the signal from the reversed video is not marginal, but rather a necessary component for instilling an understanding of the Arrow of Time. We hope this clarification of our existing results resolves the reviewer's remaining doubts.
> > >
> > > We appreciate the opportunity to clarify this point and thank the reviewer again for their valuable feedback.

---

### Official Review · Reviewer_tJ9G · 2025-07-05

**Clarity:** 3
**Significance:** 3
**Originality:** 2
**Rating:** 3
**Confidence:** 4

**Summary:**

This paper focuses on enhancing the temporal directionality (Arrow of Time) perception of MLMMs. By analyzing the limitations of existing benchmarks and models, the authors find that even SOTA MLMMs exhibit a clear insensitivity to temporal order, they often fail to distinguish between forward and reversed video playback, and may even generate identical descriptions for semantically opposite videos. To address this issue, the authors propose a GRPO-based reinforcement fine-tuning method that encourages the model to produce different interpretations for forward and reversed video frames, thereby cultivating AoT sensitivity. Additionally, they design a comprehensive benchmark to evaluate models’ AoT perception capabilities and introduce the Temporal Divergence Score (TDS) to quantify temporal sensitivity at both the benchmark and sample levels.

**Questions:**

1.	Forcing the model to learn from both forward and reversed videos may lead to the generation of physically implausible descriptions. It is recommended to include an additional set of experiments to analyze and illustrate this potential issue.
2.	It would be helpful to clarify in the paper how the questions in AoTBench were constructed, whether they were directly extracted from existing datasets or re-annotated for forward and reversed videos.
3.	If possible, providing a more detailed analysis of how different values of 𝛼 and  𝛾 affect training performance would make the paper more comprehensive and informative.

**Ethical Concerns:**

["NO or VERY MINOR ethics concerns only"]

**Limitations:**

Yes

**Quality:**

2

**Strengths And Weaknesses:**

Strengths:

1.	The reverse reward mechanism in ArrowRL is an innovative design that effectively leverages the natural contrastive signal of video temporal directionality. By employing reinforcement learning, it enhances the model’s sensitivity to the Arrow of Time.
2.	In addition to proposing a novel method, the paper also develops a dedicated benchmark (AoTBench) and conducts a systematic analysis of several existing benchmarks, providing extensive experimental data.

Weaknesses:
1.	The Reverse Reward mechanism may be overly simplistic and brute-force in nature. By forcing the model to distinguish reversed videos, it risks encouraging learning of superficial temporal cues rather than a true understanding of event dynamics. For instance, it may fail to differentiate between physically irreversible actions (e.g., glass shattering) and reversible ones (e.g., opening/closing a door). Specifically, while the method is practical and effective for reversible actions, it might lead the model to generate physically implausible descriptions for irreversible actions, as the reward function does not explicitly model physical constraints. Moreover, for cyclic or repetitive events, the Reverse Reward could have adverse effects.
2.	Were the questions in AoTBench directly extracted from existing benchmarks, or were the forward and reversed versions manually re-annotated based on videos from those datasets? If it’s the former, how is temporal sensitivity ensured? Wouldn’t that essentially mean the new benchmark is just a collection of previously difficult samples, rather than a dedicated test of temporal directionality?
3.	The paper only discusses the impact of extreme values of 𝛼 and 𝛾 (i.e., 0 and 1) on training performance, but lacks analysis of intermediate values (e.g., 0.5), which could provide more insight into the trade-offs and sensitivity of the proposed method.

---

> ### Author Rebuttal · Authors · 2025-07-29
>
> We thank reviewer tJ9G for the helpful comments and for providing thoughtful feedback on our work.
>
> ---
> **1. Reverse Reward Mechanism**
>
> > Forcing the model to learn from both forward and reversed videos may lead to the generation of physically implausible descriptions. It is recommended to include an additional set of experiments to analyze and illustrate this potential issue.
>
> We agree that applying a reverse reward indiscriminately would be problematic, and emphasize that ArrowRL was designed specifically with this nuance in mind. As noted in *Ln 207-214*, ArrowRL incorporates two key mechanisms—high-temporality data curation and dynamic weighting—to ensure that the model learns a true understanding of event dynamics rather than superficial cues.
>
> **High-temporality Data Curation**. First, at the data level, a portion of our training data is sourced from RTime [14], a benchmark of high-temporality videos where the reversed version is both physically plausible and semantically distinct from the forward version (*Ln 251-253*). By training on these vetted (manually verified to demonstrate high temporality) videos, ArrowRL learns the core patterns of reversible, meaningful actions without being confused by inapplicable cases.
> As suggested, we ran an additional experiment to isolate the effect of our data selection strategy. We train a model only for the captioning task, where the high-quality RTime data was replaced by uncurated videos from LLaVA-Video-178K [73]. As the table shows, the model trained on curated, high-temporality data performs better, underscoring the importance of our data selection strategy, and (we think) consistent with the reviewer’s expectation.
>
> | Model                          | AoTBench Acc. (%) |
> |-----------------------------|:-----------------:|
> | Base LMM                      |       56.2        |
> | w/ ArrowRL (RTime Cap.)            |       **60.4**        |
> | w/ ArrowRL (LLaVA-Video-178K Cap.) |       57.7        |
>
> **Dynamic reward weighting**. Next, to ensure scalability beyond curated data like RTime, we also train on the large, uncurated LLaVA-Video-178K dataset for open-ended QA (*Ln 248-250*), where irreversible events are likely to appear. Our dynamic reward weighting is designed to handle precisely these cases. Consider the glass shattering example the reviewer mentioned: since the reversed action is physically impossible, a model will naturally struggle to describe it. In fact, as we see across many examples, LMMs tend to default to descriptions that are semantically very close to the forward ground truth in such cases. Our mechanism detects this by calculating a high Similarity$(\tilde o, o^*)$ that surpasses a threshold 𝛾. Therefore, if the video is not meaningfully reversible, ArrowRL will "turn off" the reverse reward for that specific example by setting its weight to zero. This ensures our model is not forced into the task of creating a plausible description for an implausible event. The importance of our dynamic weighting is already ablated in *Table 2* (row +ArrowRL (𝛾=1)), which shows that removing this dynamic weighting hurts performance.
>
> We believe this two-part strategy ensures that ArrowRL is a nuanced and targeted tool, not a brute-force mechanism, and will clarify this further in the final version of the paper.
>
> ---
>
> **2. Clarification of AoTBench Construction**
>
> > It would be helpful to clarify in the paper how the questions in AoTBench were constructed, whether they were directly extracted from existing datasets or re-annotated for forward and reversed videos.
>
> We designed AoTBench to be a dedicated test of LMM temporal sensitivity, rather than a generic collection of difficult samples. As detailed in *Sec. 3.3* and *Supp. Sec. B.3*, AoTBench is a multi-faceted benchmark where each component is designed to probe this specific capability.
>
> The first two tasks, sequence direction classification & directional caption matching, are inherently temporal. For these, we use videos from existing datasets and design questions that require the model to classify a video's playback direction or match it to a corresponding forward/reverse caption (*Table 4* & *Fig. 9*). These tasks cannot be solved without understanding temporal directionality.
>
> For the third task, AoT-sensitive VQA, we compose data samples from 8 existing video benchmarks. Rather than pulling generically "hard" questions from existing benchmarks, we develop a more principled approach. We introduce the Temporal Divergence Score (TDS), a metric that precisely quantifies how critical the video's AoT is to arriving at the correct answer (*Sec. 3.1*). TDS allows us to analyze samples from different benchmarks and select only those that require true temporal perception. This principled filtering guarantees our VQA subset is a true test of temporal perception, not just a collection of unrelated hard cases, as illustrated by the examples in *Fig. 6 & 8*.
>
> ---
>
> **3. Different values of α & 𝛾**
>
> > If possible, providing a more detailed analysis of how different values of 𝛼 and 𝛾 affect training performance would make the paper more comprehensive and informative.
>
> We appreciate the suggestion and have conducted additional experiments on the hyperparameters 𝛼 (reverse reward weight) and 𝛾 (dynamic weighting threshold), building upon the ablations already in *Table 2*.
>
> **Analysis of Reverse Reward Weight (𝛼)**. Our main paper reports results for our chosen value of 𝛼 = 0.5 and includes an ablation of 𝛼 = 0 (disabling the reverse reward). As requested, we have now evaluated a fuller range of values. The results below show the performance across different weights.
>
> | Setting        | AoTBench Acc. (%) |
> |----------------|:-------------------:|
> | Base LMM       | 56.2              |
> | 𝛼 = 0          | 55.6              |
> | 𝛼 = 0.25       | **61.3**              |
> | 𝛼 = 0.5        | 60.7              |
> | 𝛼 = 0.75       | 61.1              |
> | 𝛼 = 1.0        | 54.0              |
>
> These results confirm that the reverse reward is a critical component of our method. Performance is poor at the extremes (𝛼 = 0 and 𝛼 = 1.0), but strong and stable across the range of [0.25, 0.75]. This detailed analysis identifies 𝛼 = 0.25 as the optimal weight, and we will update our model and paper accordingly.
>
> **Analysis of Dynamic Weighting Threshold (𝛾)**. Our main paper reports results for our chosen value of 𝛾 = 0.5 and ablates the case with no thresholding (𝛾 = 1). We have now tested additional intermediate values as suggested. Note that 𝛾 = 0 is an invalid setting as it would filter out all training data. The extended analysis is below.
>
> | Setting        | AoTBench Acc. (%) |
> |----------------|:-------------------:|
> | Base LMM       | 56.2              |
> | 𝛾 = 1       | 58.3             |
> | 𝛾 = 0.75        | 61.1              |
> | 𝛾 = 0.5       | 60.7              |
> | 𝛾 = 0.25        | **61.3**              |
>
> This analysis confirms our dynamic weighting is a critical component, as removing it (𝛾 = 1) greatly degrades performance. The method is also robust, with stable and strong results across the [0.25, 0.75] range. This detailed analysis identifies 𝛾 = 0.25 as the optimal weight. We thank the reviewer for this valuable suggestion and will update our model and paper accordingly.

---

> ### Comment · Area_Chair_TqnZ · 2025-08-06
> **Author-reviewer Discussion**
>
> Dear Reviewer tJ9G,
>
> The system noticed that you haven't post any discussion with the authors yet. As per the review guidelines, reviewers are expected respond to authors’ rebuttal, ask further questions (if any) and listen to answers to help clarify remaining issues before submitting Mandatory Acknowledgement.  Please engage with the authors now to offer your feedback on their rebuttal.
>
> Thank you!
>
> AC, NeurIPS 2025

---

### Comment · Area_Chair_TqnZ · 2025-08-03
**Reviewer-Author Discussion**

Dear Reviewers,

The discussion period with the authors will remain open until August 6th (AoE). Please take the time to read and acknowledge the authors' rebuttals, and post any follow-up questions or comments you may have.

Best regards,
AC

---

### Note · Authors · 2025-08-11

We sincerely thank all reviewers and the Area Chair for their time and valuable feedback, which has helped us strengthen our paper.

We were pleased to have constructive discussions that resolved the concerns of **Reviewers GqEN (Accept)** and **6fDT (Borderline Accept)**.

**Reviewer mxwr indicated they'd raise their score**, but had the lingering question about whether our method's impact is "marginal." As our final reply clarified, the ablation study in Table 2 of the main paper provides direct evidence against this: removing our reverse-time signal (α=0) results in a significant performance drop compared to our full method (55.6% v.s. 60.7%).

Our rebuttal also provided experiments and clarifications for the initial concerns raised by **Reviewer tJ9G (Borderline Reject)**. Specifically, we demonstrated that our reward mechanism is nuanced, not simplistic, via its two-part design and an experiment on data curation; we justified our benchmark's principled construction; and, as requested, we provided a full hyperparameter sensitivity analysis that further validated our method's effectiveness. Although we did not receive a reply from this reviewer during the discussion phase, we hope our clarifications have addressed their concerns.

We are grateful for the opportunity to improve our paper through this process. We thank everyone again for their time and thoughtful consideration of our work.

---

### Decision · Program_Chairs · 2025-09-17

**Decision:**

Accept (poster)

**Comment:**

This submission addresses the lack of temporal directionality (Arrow of Time) sensitivity in large multimodal models. The proposed ArrowRL method combines a reverse reward with curated data and dynamic weighting, and AoTBench provides a principled benchmark for evaluating AoT sensitivity.

The paper received mixed reviews. Reviewer GqEN recommended acceptance, highlighting methodological soundness and the broader value of AoTBench. Reviewer 6fDT gave a borderline accept, finding the rebuttal convincing on issues of forward/reverse response construction and generalization. Reviewer mxwr remained skeptical, concerning that the improvements may be marginal and potentially achievable without RL; although their score increased after rebuttal, it remained borderline negative. Reviewer tJ9G raised concerns about the reward mechanism, benchmark construction, and hyperparameter sensitivity, but did not re-engage in the discussion phase. After examining the rebuttal, the AC believes these concerns are properly addressed by the authors’ additional experiments.

Overall, the authors and most pf the reviewers engaged actively and constructively during the discussion. The added hyperparameter analyses, control experiments showing that poor AoT performance is not simply due to OOD effects, and expanded evaluations on broader benchmarks strengthen the case for the method. The rebuttal also clarified AoTBench’s principled design through the Temporal Divergence Score. While some conceptual issues remain (e.g., handling irreversible events in language, or the necessity of RL compared to simpler methods), these are acknowledged and framed as promising directions for future work.

Given these contributions and discussions, and despite lingering doubts about the magnitude of impact, the AC recommends acceptance. The paper is technically solid, makes a timely contribution, and provides resources that are likely to spur further research into temporal reasoning for LMMs.